# Exploring High-Order Self-Similarity for Video Understanding

## Abstract

Space-time self-similarity (STSS), which captures visual correspondences across frames, provides an effective way to represent temporal dynamics for video understanding. In this work, we explore higher-order STSS and demonstrate how STSS at different orders reveals distinct aspects of these dynamics. We then introduce the Multi-Order Self-Similarity (MOSS) module, a lightweight neural module designed to learn and integrate multi-order STSS features. It can be applied to video understanding tasks to enhance motion modeling capabilities while consuming only marginal computation cost and memory usage. Extensive experiments on motion-centric action recognition benchmarks, i.e., Something-Something V1 & V2, Diving48, and FineGym, our method achieves new state-of-the-art results, presenting the best memory-accuracy trade-off compared to existing approaches. The source code and checkpoints of our model will be publicly available.

## 1 Introduction

The real world is dynamic, not static. The most prominent characteristic that distinguishes videos from images lies in the presence of such temporal dynamics, *i.e.*, changes of visual patterns over time. Without a proper grasp of those features, *e.g.*, motion information, video understanding models often become biased toward static contextual cues, limiting their generalization in out-of-context scenarios (Bae et al., 2023; Choi et al., 2019; Chung et al., 2022; Li et al., 2018).

Temporal dynamics in general can be represented as structural patterns of how visual elements interact to each other in space and time. While the most popular and explicit form of it would be motion fields or optical flows (Dosovitskiy et al., 2015; Ng et al., 2018; Sun et al., 2018; Teed & Deng, 2020), the seminar work by Shechtman & Irani (2005; 2007) has shown that the space-time self-similarity (STSS), *i.e.*, a correlation volume over a local window of a video in space and time, effectively

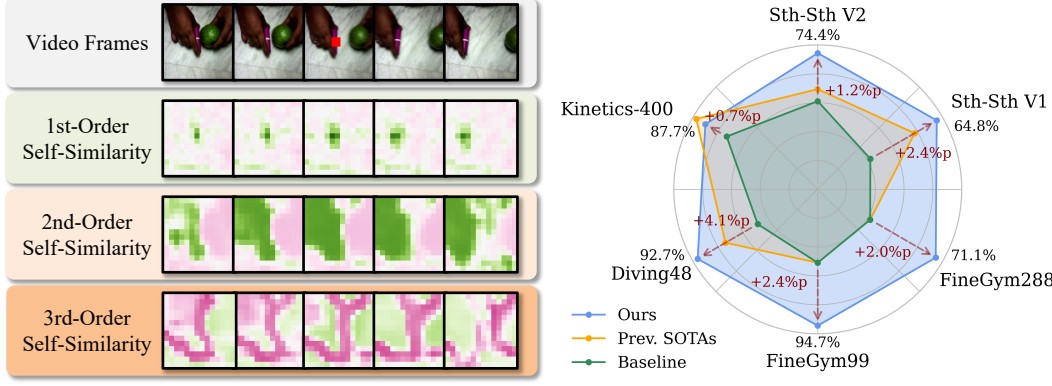

(a) **STSS maps across different orders.**     (b) **Action recognition performance.**

Figure 1: **High-order space-time self-similarities (STSS) for effective video understanding.** (a) Given a red query, the 1st-, 2nd-, and 3rd-order STSS effectively identify *motion flows, motion segments, and the layout of motion segments*, respectively. (b) We leverage the high-order STSS to capture diverse aspects of spatio-temporal dynamics in videos, resulting in significant performance improvements across various action recognition benchmarks.

reveals temporal dynamics and suppressing irrelevant appearance variations. Recent studies (Bian et al., 2022; Kim et al., 2021; Kwon et al., 2021; Son, 2022; Wang et al., 2020; Wu et al., 2023) also demonstrate that learning self-similarity features on latent space-time feature maps enables neural networks to understand motion in videos better, improving performance in action recognition.

In this work, we explore higher-order self-similarities, *e.g.*, *self-similarity of self-similarity* in space and time as the 2nd-order STSS, and investigate what kinds of distinct temporal dynamics emerge. We are motivated by the fact that the role of self-similarity operation is to reveal the structure of correlation patterns (Fig. 1a); given a base feature map describing appearance for each position in space and time, the conventional 1st-order STSS computes similarities of appearances, revealing *motion flows*, *e.g.*, the leftward translation of the queried pen across frames (2nd row). In the same vein, given the 1st-order STSS map describing motion flows, the 2nd-order STSS compute similarities of motion, recognizing *motion segments*; the 2nd-order STSS maps highlight regions of both the hand and pen that share similar motion patterns regardless of their distinct appearances (3rd row). The 3rd-order STSS further extends these correlation patterns by capturing similar motion segments from the 2nd-order STSS features, effectively identifying the *layout of motion segments* (4th row). This hierarchical progression to higher-order STSS provides useful cues for the comprehensive video analysis in complex scenarios.

From these insights, we design a novel neural module, dubbed MOSS (**M**ulti-**O**rder **S**elf-**S**imilarity), that learns distinct representations of STSSs at diverse orders and integrates them into holistic motion features. The proposed module is lightweight and can be inserted into existing video architectures to enhance video representations. We demonstrate the effectiveness of the MOSS module by incorporating it with a ladder side tuning (LST) framework (Jiang et al., 2024; Qing et al., 2023; Sung et al., 2022; Yao et al., 2023). Specifically, we freeze the pre-trained image encoder and train a lightweight temporal encoder in parallel, taking intermediate features from the image encoder as input. By inserting the MOSS module between the two encoders, we allow the temporal encoder to effectively utilize both the visual and the multi-order STSS features for motion-enhanced video representation learning. We evaluate our method on diverse action recognition benchmarks, *i.e.*, Kinetics-400, Something-Something V1 & V2, Diving48, and FineGym, demonstrating significant performance improvements (Fig. 1b), introducing marginal computation and memory overhead, leading to favorable memory-accuracy trade-off (Tab. 3).

Our contributions are summarized as:

- We provide an in-depth analysis of high-order space-time self-similarities and discover that each order exhibits unique and complementary temporal dynamics.

- We propose MOSS, a novel lightweight neural module that learns integrated STSS features at multiple orders for comprehensive temporal understanding.

- We propose a memory-efficient image-to-video transfer method achieving strong performance on action recognition with favorable memory-accuracy trade-off.

## 2 RELATED WORK

**Self-Similarity in Video Understanding.** The pioneering work by Shechtman & Irani (2005; 2007) has shown that the self-similarity, *i.e.*, a correlation over a local window of an image or a video in space and time, effectively reveals structural layouts and suppresses irrelevant appearance variations. Based on this, Junejo et al. (2010; 2008) propose robust temporal self-similarity descriptors that recognize human actions under view changes. Recently, several methods (Kwon et al., 2020; 2021; Wang et al., 2020) employ self-similarities within a video clip for learning motion features. CorrNet (Wang et al., 2020) and MSNet (Kwon et al., 2020) compute spatial cross-similarities between adjacent frames to obtain short-term motions, and SELFY (Kwon et al., 2021) proposes a neural module that learns STSS representations as bi-directional motion features. Wu et al. (2023) extend this work by combining STSS with frame-wise differences to capture richer temporal dynamics in videos. As self-attention mechanism rises, various transformer architectures (Arnab et al., 2021; Fan et al., 2021; Li et al., 2023a;b; Liu et al., 2022) have been proposed for video understanding. Although these architectures do not explicitly leverage self-similarities, they adopt space-time correlations in an attention-based manner. Some methods (Kim et al., 2021; 2024) improve the self-attention mechanism to leverage STSS features for better video representation learning. However, none of these methods explore the high-order STSS, *i.e.*, self-similarity of self-similarity in space-time. To

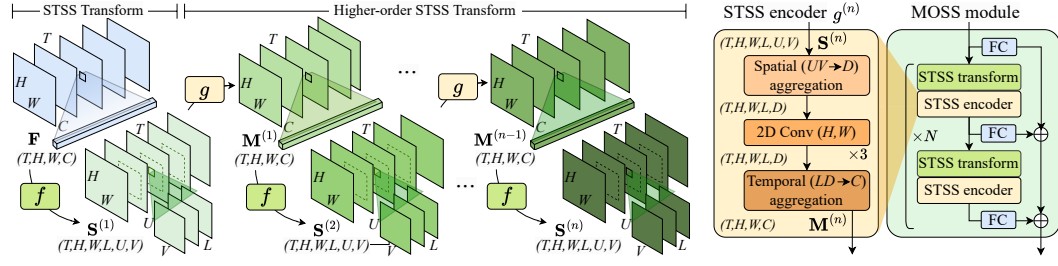

(a) STSS & Higher-order STSS Transformation          (b) MOSS module

Figure 2: **High-order STSS transformation & Multi-Order Self-Similarity (MOSS) module.** (a) depicts a recursive process for high-order STSS transformation. (b) illustrates the overall process of the MOSS for learning multi-order STSS representations.

the best of our knowledge, our work is the first to introduce high-order STSS and show their unique contributions in describing temporal dynamics in videos.

**Efficient Image-to-Video Transfer.** With the advance in large vision foundation models (Cherti et al., 2023; Oquab et al., 2023; Radford et al., 2021; Sun et al., 2023; 2024; Bardes et al., 2023; Assran et al., 2025), efficient image-to-video transfer methods (Lin et al., 2022; Pan et al., 2022; Park et al., 2023; Qing et al., 2023; Yang et al., 2023; Yao et al., 2023; Wang et al., 2024; Liu et al., 2024a) have increasely gained attention as alternatives to end-to-end finetuning (Arnab et al., 2021; Bertasius et al., 2021; Fan et al., 2021; Kim et al., 2024; Li et al., 2023a;b; 2022b; Liu et al., 2022; Wu et al., 2023; Yan et al., 2022). Current efficient image-to-video transfer approaches can be categorized into two streams. The first stream adopts Parameter-Efficient Fine-Tuning (PEFT) (Houlsby et al., 2019; Hu et al., 2021) by integrating lightweight spatio-temporal adapters into frozen image encoders (Pan et al., 2022; Park et al., 2023; Yang et al., 2023; Wang et al., 2024; Liu et al., 2024a). However, these methods still require excessive memory overhead for gradient backpropagation through the encoder during training. Meanwhile, the second stream focuses on memory-efficient finetuning by adopting the Ladder Side-Tuning (LST) framework (Sung et al., 2022; Jiang et al., 2024) in NLP, which processes features in parallel to the frozen encoder through lightweight side networks (Lin et al., 2022; Qing et al., 2023; Yao et al., 2023). In specific, Qing et al. (2023) design two CNN-based lightweight networks that learns temporal dynamics and integrates spatial and temporal features. Yao et al. (2023) introduce Side4Video, an lightweight temporal encoder combining temporal convolutions and transformers, facilitating the efficient training of video models using large-scale ViT-E/14 backbone (Sun et al., 2023). In this work, we enhance temporal modeling capabilities within the LST paradigm by integrating the proposed MOSS module between the frozen image encoder and side network, enabling memory-efficient image-to-video transfer through high-order STSS features.

## 3 OUR APPROACH

We first revisit the concept of *space-time self-similarity* (STSS), then extend it to higher orders, and discuss the distinct information captured at each order. We then introduce our MOSS module that learns to exploit the distinct STSS representations across the diverse orders and integrate them into holistic motion features. Finally, we describe our video model that incorporates MOSS with a ladder side tuning model (Yao et al., 2023) for memory-efficient image-to-video transfer.

### 3.1 REVISITING SPACE-TIME SELF-SIMILARITY (STSS)

**STSS Transformation.** Self-similarity (Shechtman & Irani, 2007) reveals geometric structures of correlations between visual entities while suppressing their visual content, allowing us to understand relational patterns in visual data. In the video domain, STSS computes pair-wise correlations between a query and its local spatio-temporal neighbors, describing spatio-temporal dynamics of the query across frames. We define an STSS transformation function $f$ that maps input feature maps to a 6D tensor as,

$$f : \mathbb{R}^{T \times H \times W \times C} \to \mathbb{R}^{T \times H \times W \times L \times U \times V}, \tag{1}$$

where $(T, H, W)$ are the spatio-temporal dimensions, and $(L, U, V)$ denote the size of the local spatio-temporal window. Given input feature maps of $T$ frames $\mathbf{F} \in \mathbb{R}^{T \times H \times W \times C}$, each element of

the STSS tensor $\mathbf{S} = f(\mathbf{F}) \in \mathbb{R}^{T \times H \times W \times L \times U \times V}$ is computed as,

$$\mathbf{S}_{t,h,w,l,u,v} = \phi(\mathbf{F}_{t,h,w}, \mathbf{F}_{t+l,h+u,w+v}), \qquad (2)$$

where $(t, h, w)$ is 3D coordinates of a query and $(l, u, v)$ is an offset of local spatio-temporal window of the query, where $(l, u, v) \in [-\lfloor\frac{L}{2}\rfloor, \lfloor\frac{L}{2}\rfloor] \times [-\lfloor\frac{U}{2}\rfloor, \lfloor\frac{U}{2}\rfloor] \times [-\lfloor\frac{V}{2}\rfloor, \lfloor\frac{V}{2}\rfloor]$. The function $\phi$ computes the similarity, *e.g.*, cosine similarity, between two feature vectors.

**Characteristics of STSS.** The STSS tensor $\mathbf{S}$ effectively captures appearance-based correspondences across different frames, presenting diverse temporal information throughout the video sequence. For $l = 0$, it is spatial self-similarity (Shechtman & Irani, 2007) showing object layouts or similar objects within the same frame. For $l \neq 0$, it becomes spatial cross-similarity between two different frames, presenting a displacement map of the query, commonly used in motion feature learning (Kwon et al., 2020; Wang et al., 2020) or optical flow estimation (Bian et al., 2022; Ng et al., 2018; Sun et al., 2018; Teed & Deng, 2020). By connecting the regions across the $L$ frames, the tensor turns out to reveal the *motion flows* of the query over time.

### 3.2 GENERALIZATION TO HIGHER-ORDER STSS

**High-Order STSS Transformation.** Unlike the conventional STSS, higher-order STSS explores the *similarity of similarity* patterns themselves, providing a deeper understanding of motion dynamics. However, recursively applying $f$ is impractical since the tensor dimension increases exponentially. To address this, we introduce an STSS encoding function $g : \mathbb{R}^{T \times H \times W \times L \times U \times V} \to \mathbb{R}^{T \times H \times W \times C}$, which abstracts features from the STSS tensor while mapping the high-dimensional tensor to the original feature space. Note that $g$ can be an arbitrary function including vectorization, pooling operations, parametrized learnable encoders, or their compositions. By composing $f$ and $g$, we can define a recursive process for computing higher-order STSS tensors while keeping the feature dimensions consistent (Fig. 2a). Considering the original STSS is presented as the 1st-order STSS tensor $\mathbf{S}^{(1)}$, we define the $n$-th order STSS tensor $\mathbf{S}^{(n)}$ recursively as,

$$\mathbf{S}^{(n)} = \begin{cases} f(\mathbf{F}), & \text{if } n = 1 \\ f \circ g\left(\mathbf{S}^{(n-1)}\right), & \text{if } n \geq 2. \end{cases} \qquad (3)$$

**Advantages and Key Features of High-Order STSS.** Higher-order STSSs play distinct roles compared to the 1st-order STSS in understanding spatio-temporal dynamics. While the 1st-order STSS reveals basic motion flows (*e.g.* existence of motion, directions) by computing similarities based on appearance across frames, 2nd-order STSS computes motion-based similarities and reveals regions with coherent motion patterns, akin to *motion segments*. Since pixels within the same object share similar motion flows, these motion segments effectively highlight moving objects and their motion trajectories from complex scenes. This distinct nature of the 2nd-order STSS provides crucial complementary temporal cues in scenarios where the 1st-order STSS fails; the 1st-order STSS struggles to distinguish pure motion of the query from motions of other visually similar objects (2nd row, Fig. 3), whereas, the 2nd-order STSS successfully separates the query's motion from other visually similar object while highlighting other objects with similar motion patterns (3rd row, Fig. 3).

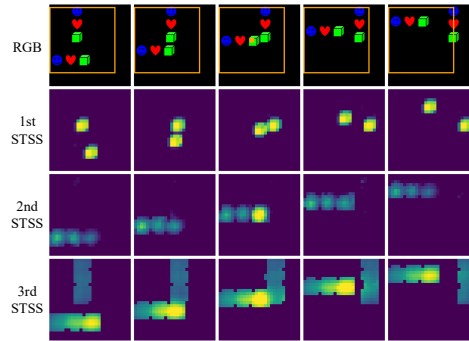

Figure 3: **STSS map visualizations on a toy video clip**. From top to bottom, we visualize RGB frames and 1st-, 2nd-, and 3rd-order STSS maps of the brown query by setting STSS encoding function $g$ as vectorization over $(L, U, V)$ dimensions. The STSS maps here progressively capture different temporal dynamics: motion flow, motion segments, and overall motion layouts.

The 3rd-order STSS further extends this hierarchy by computing similarities based on motion segments, demonstrating the *overall layouts of these segments*. Unlike the 2nd-order STSS that identifies individual motion segments, the 3rd-order STSS captures how these segments interact with each other, revealing the overall motion patterns (4th row, Fig. 3). This can be beneficial for understanding complex actions that involve multiple simultaneous motions (*e.g.* group actions). Interestingly, despite

the potential of modeling group dynamics, we observe that our trained models, in practice, learn to leverage the 3rd-order STSS to highlight motion boundaries of multiple motion segments where distinct motion patterns naturally emerge near (dis)occlusions, as opposed to continuous motion regions (4th row, Fig. 1a). This progression from 1st-order STSS to higher-order STSS unveils diverse aspects of motion dynamics in videos for comprehensive video understanding.

While our framework theoretically supports higher-order computations ($n \geq 4$), our empirical analysis suggests that STSS beyond 3rd-order do not provide significant benefits for action recognition tasks.

### 3.3 Learning Multi-Order STSS Representations

We here introduce MOSS (Multi-Order Self-Similarity) module, a lightweight neural module that transforms multi-order STSS tensors into neural motion features. We first explain our STSS encoder $g$ that effectively exploits structural patterns of the STSS tensor at each order and then describe MOSS that combines multi-order STSS features into a deeper motion representation.

**STSS Encoder.** To obtain the $n$-th order STSS representation $\mathbf{M}^{(n)}$, we express the computation as,

$$\mathbf{M}^{(n)} = g^{(n)}\left(\mathbf{S}^{(n)}\right),\tag{4}$$

where we employ an independent STSS encoder $g^{(i)}$ for each order. We design $g$ in late-fusion manner, *i.e.*, encode spatial structures first then fuse temporal information (Kwon et al., 2021), as illustrated in Fig. 2b. We first transform the structural patterns of each spatial similarity map across $L$ frames into $D$-dimensional vector by flattening the $(U, V)$ dimensions and applying a fully connected layer, resulting in a tensor of size $\mathbb{R}^{T \times H \times W \times L \times D}$. While the previous methods (Kwon et al., 2021; Wu et al., 2023) used a series of 2D convolutions for $(U, V)$ extraction, we found that a simple linear layer achieves competitive performance while reducing memory overhead. Next, we refine the spatial similarity features by applying a series of 2D convolutions over $(H, W)$ dimensions. Each convolution block consists of $\texttt{Conv2d} - \texttt{BatchNorm} - \texttt{GeLU}$, maintaining $D$ channels. Finally, we concatenate $L$ refined similarity features along the channel dimension and apply a fully connected layer to integrate features across temporal offsets, resulting in a tensor of size $\mathbb{R}^{T \times H \times W \times C}$.

**MOSS Module.** The final output feature maps are obtained by combining the multi-order STSS feature maps with the original visual feature maps as,

$$\texttt{MOSS}(\mathbf{F}) = \texttt{FC}\left(\mathbf{F}\right) + \sum_{n=1}^{N} \texttt{FC}\left(\mathbf{M}^{(n)}\right).\tag{5}$$

This combination allows our model to leverage both the original visual features and the diverse motion patterns captured by multi-order STSS features.

### 3.4 Video Architecture

**Overall Framework.** The proposed module is generic so it can be inserted into existing video architectures. Here, we integrate the MOSS module into a ladder side tuning architecture to achieve memory-efficient image-to-video transfer learning, illustrated in (Fig. 4). Our framework consists of a pretrained spatial encoder, a lightweight temporal encoder, and our MOSS module that bridges the two networks. The image encoder is frozen, and intermediate features are extracted from different layers of the encoder. MOSS module then computes multi-order STSSs on these visual features and then transforms them into motion features. Finally, the temporal encoder takes the motion-augmented visual features and outputs motion-centric video representations.

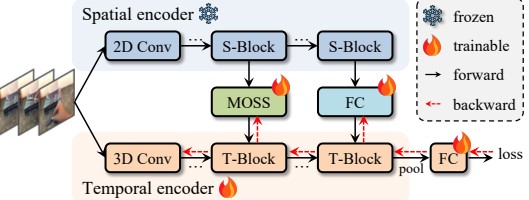

Figure 4: **Overall video architecture**.

**Spatial Encoder.** We use CLIP-pretrained ViT as the spatial encoder. Given an input video $\mathbf{X} \in \mathbb{R}^{T \times H' \times W' \times 3}$, let us denote a sequence of input token embeddings at the $t$-th frame as $\bar{\boldsymbol{F}}_t^0 = [\boldsymbol{f}_{\text{cls}}^0; \boldsymbol{F}_t^0] \in \mathbb{R}^{(HW+1) \times C}$ where $\boldsymbol{f}_{\text{cls}}^0$ and $\boldsymbol{F}_t^0$ are class

Table 1: **Results on motion-centric video benchmarks.** [†] trained with text supervision. * reproduced by our setup. "Input" indicates # frames×# crops×#clips.

(a) Something-Something V1 & V2.

| method | backbone | pre-train | input | TFLOPs | SSV1 top1 | SSV1 top5 | SSV2 top1 | SSV2 top5 |
|---|---|---|---|---|---|---|---|---|
| *Full finetuning* | | | | | | | | |
| ViViT | L/16×2 FE | IN21K,K400 | 32×12 | 1.0×12 | - | - | 65.9 | 89.9 |
| UniFormerV2 | ViT-L/14 | CLIP400M | 32×3 | 1.73×3 | 62.7 | 88.0 | 73.0 | 94.5 |
| ATM | ViT-L/14 | CLIP400M | 16×6 | 0.84×6 | 64.0 | 88.0 | 73.5 | 93.7 |
| *Frozen backbone* | | | | | | | | |
| V-JEPA | ViT-H/16 | VM2M | 16×6 | - | - | - | 74.3 | - |
| V-JEPA 2 | ViT-H/16 | VM22M | 16×6 | - | - | - | 74.0 | - |
| M²CLIP[†] | ViT-B/16 | CLIP400M | 16×12 | - | - | - | 67.3 | - |
| OmniCLIP[†] | ViT-B/16 | CLIP400M | 32×3 | 0.8×3 | - | - | 69.1 | 91.8 |
| EVL | ViT-L/14 | CLIP400M | 32×3 | 3.21×3 | - | - | 66.7 | - |
| ST-Adapter | ViT-L/14 | CLIP400M | 32×3 | 2.75×3 | - | - | 72.3 | 93.9 |
| DualPath | ViT-L/14 | CLIP400M | 48×3 | 0.72×3 | - | - | 72.2 | 93.7 |
| AIM | ViT-L/14 | CLIP400M | 32×3 | 3.84×3 | - | - | 70.6 | 92.7 |
| DiST[†] | ViT-L/14 | CLIP400M | 32×3 | 2.83×3 | - | - | 73.1 | 93.2 |
| MoTED[†] | ViT-L/14 | CLIP400M | 32×3 | 2.89×3 | - | - | 73.8 | 93.8 |
| Qian *et al.* | ViT-L/14 | CLIP400M | 32×3 | 1.69×3 | - | - | 73.6 | 94.3 |
| Side4Video | ViT-B/16 | CLIP400M | 16×6 | 0.36×6 | 60.7 | 86.0 | 71.5 | 92.8 |
| Side4Video | ViT-L/14 | CLIP400M | 16×6 | 1.74×6 | 62.4 | 88.1 | 73.2 | 93.9 |
| Side4Video | ViT-E/14 | Merged-2B | 16×6 | 15.96×6 | 67.3 | 88.8 | 75.2 | 94.0 |
| MOSS-B (ours) | ViT-B/16 | CLIP400M | 8×6 | 0.18×6 | 61.0 | 86.1 | 71.4 | 93.0 |
| MOSS-B (ours) | ViT-B/16 | CLIP400M | 16×6 | 0.36×6 | 61.8 | 86.8 | 72.4 | 93.5 |
| MOSS-L (ours) | ViT-L/14 | CLIP400M | 8×6 | 0.83×6 | 63.6 | 87.9 | 73.1 | 93.3 |
| MOSS-L (ours) | ViT-L/14 | CLIP400M | 16×6 | 1.67×6 | 64.8 | 89.0 | 74.4 | 94.4 |
| MOSS-L (ours) | ViT-L/14 | Merged-2B | 16×6 | 1.67×6 | **67.3** | **89.8** | **75.3** | **94.5** |

(b) Diving48.

| method | top1 |
|---|---|
| TimeSformer-HR | 78.0 |
| TimeSformer-L | 81.0 |
| V-JEPA | 87.9 |
| ORViT | 88.0 |
| StructViT-B-4-1 | 88.3 |
| Side4Video-B* | 88.6 |
| V-JEPA 2 | 89.8 |
| AIM ViT-L | 90.6 |
| Video-FocalNet-B | 90.8 |
| MOSS-B (ours) | 91.2 |
| MOSS-L (ours) | **92.7** |

(c) FineGym

| method | gym99 | gym288 |
|---|---|---|
| TSM | 70.6 | 34.8 |
| TSM$_{two-stream}$ | 81.2 | 46.5 |
| RSANet | 86.4 | 50.9 |
| StructViT-B-4-1 | 89.5 | 54.2 |
| TQN | 90.6 | 61.9 |
| VT-CE | 91.4 | 62.6 |
| Side4Video-B* | 92.3 | 69.1 |
| MOSS-B (ours) | 93.9 | 70.2 |
| MOSS-L (ours) | **94.7** | **71.1** |

and visual embeddings, respectively. The intermediate features after the $i$-th transformer block at the $t$-th frame are computed as,

$$\bar{\boldsymbol{F}}_t^i = \texttt{S-Block}_l(\bar{\boldsymbol{F}}_t^{i-1}), \quad i = 1, \ldots, N^{\text{s}}. \tag{6}$$

We collect the visual features across all frames $\mathbf{F}^i = \{\boldsymbol{F}_1^i, \cdots, \boldsymbol{F}_T^i\} \in \mathbb{R}^{T \times H \times W \times C}$ and pass them to the subsequent MOSS module and the temporal encoder.

**MOSS Module Integration.** We apply our MOSS module at the $k$-th layer and compute STSS-augmented features as,

$$\mathbf{G}^i = \begin{cases} \texttt{MOSS}(\mathbf{F}^i) & \text{if } i = k \\ \texttt{FC}(\mathbf{F}^i) & \text{otherwise.} \end{cases} \tag{7}$$

**Temporal Encoder.** We adopt Side4Video (Yao et al., 2023) as the temporal encoder. This encoder first tokenizes the input video $\mathbf{X}$ to video embeddings $\mathbf{Y} \in \mathbb{R}^{T \times H \times W \times C}$ and then processes them through a sequence of `T-Block`s, where each block consists of temporal convolution, spatial self-attention with TokShift (Zhang et al., 2021b), and MLP layers. At each block, before processing the features, we combine the video features with $\mathbf{G}^i$ as,

$$\mathbf{Y}^i = \texttt{T-Block}_l(\mathbf{Y}^{i-1} + \mathbf{G}^i), \quad i = 1, \ldots, N^{\text{t}}, \tag{8}$$

$$\texttt{T-Block}(\cdot) = \texttt{MLP}(\texttt{TS-Attn}(\texttt{T-Conv}(\cdot))), \tag{9}$$

We apply global average pooling after the final block and pass the feature to a action classifer.

## 4 EXPERIMENTS

### 4.1 EXPERIMENTAL SETUP

**Datasets.** *Something-Something-V1 & V2* (Goyal et al., 2017; Mahdisoltani et al., 2018) contain 108k and 220K video clips, respectively, focusing on fine-grained actions. *Diving48* (Li et al., 2018) is a human diving action dataset consisting of 18K videos with 48 classes. *FineGym* (Shao et al., 2020a) is a fine-grained action benchmark containing 33K gymnastics videos. These datasets emphasize temporal relationships through motion-centric action categories, where success depends on accurate modeling of complex spatio-temporal dynamics. *Kinetics-400* (Kay et al., 2017) is a large-scale video dataset with 400 action classes. We use 241K action clips available online.

Table 2: **Results on Kinetics-400**. † trained with text supervision. "Input" indicates #frames×#crops×#clips.

| method | input | TFLOPs | top1 | top5 |
|---|---|---|---|---|
| *Full finetuning* | | | | |
| ViViT-H | 32×12 | 3.98×12 | 84.9 | 95.8 |
| MTV-H | 32×12 | 3.71×12 | 85.8 | 96.6 |
| XCLIP-L† | 16×12 | 3.09×12 | 87.7 | 97.4 |
| Text4Vis-L† | 32×12 | 1.66×12 | 87.6 | 97.8 |
| ATM ViT-L | 32×12 | 1.68×12 | 88.0 | 97.6 |
| *Frozen backbone* | | | | |
| V-JEPA ViT-H | 16×6 | - | 84.5 | |
| V-JEPA 2 ViT-H | 16×6 | - | 85.3 | |
| M²CLIP ViT-B | 32×12 | 0.8×12 | 84.1 | - |
| OmniCLIP ViT-B | 8×12 | 0.1×12 | 84.1 | - |
| ST-Adapter ViT-L | 32×3 | 2.75×3 | 87.2 | 97.6 |
| AIM ViT-L | 32×3 | 3.74×3 | 87.5 | 97.7 |
| DiST ViT-L† | 32×3 | 2.83×3 | **88.0** | **97.9** |
| Side4Video-B | 32×12 | 0.72×12 | 84.2 | 96.5 |
| Side4Video-L | 16×12 | 1.74×12 | 87.0 | 97.5 |
| CLIP4Vis ViT-L† | 8×12 | 0.42×12 | 87.4 | 97.9 |
| MOSS-B (ours) | 32×12 | 0.72×12 | 85.2 | 96.8 |
| MOSS-L (ours) | 16×12 | 1.67×12 | 87.7 | 97.7 |

Table 3: **Efficiency comparison**. "FLOPs", "TP", and "Mem" indicate FLOPs (G), trainable parameters (M), and memory footprint (GB), respectively. Memory footprints are measured using batch sizes of 32 and 16 for ViT-B and ViT-L, respectively.

| scale | method | FLOPs | TP | Mem | SSV2 |
|---|---|---|---|---|---|
| B/16 | ST-Adapter | 455 | 7 | 28.8 | 67.1 |
| | AIM | 624 | 14 | 35.2 | 66.4 |
| | EVL | 512 | 89 | 17.9 | 61.0 |
| | DiST | 480 | 19 | 12.7 | 68.7 |
| | Side4Video | 528 | 21 | 18.8 | 70.2 |
| | MOSS-S (ours) | **453** | **6** | **9.9** | 70.5 |
| | MOSS-B (ours) | 538 | 22 | 21.6 | **71.1** |
| L/14 | ST-Adapter | **2062** | **20** | 51.4 | 70.0 |
| | AIM | 2877 | 50 | 64.3 | 67.6 |
| | EVL | 2411 | 350 | 33.0 | 65.1 |
| | DiST | 2130 | 32 | 18.1 | 70.8 |
| | Side4Video | 2611 | 102 | 37.0 | 71.8 |
| | MOSS-M (ours) | 2120 | 24 | **17.9** | 72.0 |
| | MOSS-L (ours) | 2500 | 82 | 36.5 | **72.9** |

**Implementation Details.** We adopt ViT-B/16 and ViT-L/14 from OpenAI-CLIP (Radford et al., 2021) as the spatial encoder for MOSS-{S,B} and MOSS-{M,L}, respectively. While MOSS-{S,M} share the same spatial encoder as MOSS-{B,L} respectively, they employ more lightweight temporal encoders for efficient video processing. We insert a single MOSS module that encodes 1st- and 2nd-order STSS features. Please refer to Sec. B for detailed model and training configurations.

## 4.2 COMPARISON TO STATE-OF-THE-ART METHODS

**Something-Something V1 & V2.** In Tab. 1a, we present the results on Something-Something V1 and V2. Using 8 input frames only, MOSS-B achieves 61.0% and 71.4% top-1 accuracies on V1 and V2, respectively, which are already comparable to Side4Video (Yao et al., 2023) with 16 frames requiring only half the computational cost. Using 16 frames, MOSS-B attains top-1 accuracies of 61.8% and 72.4% on V1 and V2, respectively, outperforming existing both adapter-based PEFT methods (Pan et al., 2022; Park et al., 2023; Yang et al., 2023; Wang et al., 2024; Liu et al., 2024a) and full finetuning methods (Arnab et al., 2021; Li et al., 2023a; Wu et al., 2023) using the larger ViT-L/14 backbone. Scaling up to MOSS-L using 16 frames, we achieve strong performances of 64.8% on V1 and 74.4% on V2, significantly surpassing all the CLIP-based methods at the same ViT-L scale and even competing with video foundation models (Bardes et al., 2023; Assran et al., 2025) using larger ViT-H backbones. Finally, we replace the spatial encoder with EVA-CLIP (Sun et al., 2023) and obtain 67.3% on V1 and 75.3% on V2, competitive to Side4Video with larger ViT-E backbone while requiring 10× fewer FLOPs. These results demonstrate the effectiveness of high-order STSSs in understanding temporal dynamics.

**Diving48 & FineGym.** We summarize the results on Diving48 and FineGym in Tabs. 1b and 1c, respectively, which contain more complex motion patterns compared to Something-Something datasets. For both benchmarks, MOSS-B outperforms all other methods (Lin et al., 2019; Kim et al., 2021; Zhang et al., 2021a; Bertasius et al., 2021; Leong et al., 2022; Herzig et al., 2022; Bardes et al., 2023; Yang et al., 2023; Yao et al., 2023; Kim et al., 2024; Assran et al., 2025); MOSS-L obtains 92.7% on Diving48, 94.7% on gym99, and 71.1% on gym288, achieving state-of-the-art with substantial margins.

**Kinetics-400.** In Tab. 2, our method also demonstrates its effectiveness on Kinetics-400, which is an appearance-centric benchmark. MOSS-B and MOSS-L achieve 85.2% and 87.7% top-1 accuracies, improving over the baseline by 1.0%p and 0.7%p, respectively, competitive to other methods (Arnab et al., 2021; Yan et al., 2022; Ma et al., 2022; Wu et al., 2023; Lin et al., 2022; Pan et al., 2022; Yang et al., 2023; Qing et al., 2023; Bardes et al., 2023; Yao et al., 2023; Wang et al., 2024; Liu et al., 2024a; Wu et al., 2024; Assran et al., 2025). This validates the generalizability of our high-order STSS features, which can be effectively leveraged across diverse video domains.

Table 4: **Ablation studies on Something-Something V1 and Diving48 dataset.** All the experiments are conducted with MOSS-S taking 8 and 32 frames input on Something-Something V1 and Diving48, respectively. "FLOPs", "TP", and "Mem" respectively indicate FLOPs (G), trainable parameters (M), and memory footprint (GB) using 8 frames. Memory footprint is measured using a batch size of 32 for a single GPU machine. Rows in gray indicate our default configurations.

(a) Effect of High-Order STSS

| $n$=1 | 2 | 3 | 4 | FLOPs | TP | Mem | SSV1 | D48 |
|---|---|---|---|---|---|---|---|---|
| | | | | 148.4 | 4.5 | 8.0 | 56.9 | 85.0 |
| ✓ | | | | 150.0 | 5.1 | 9.0 | **59.0** | **86.3** |
| | ✓ | | | 151.5 | 5.6 | 9.9 | 58.7 | 86.1 |
| | | ✓ | | 152.9 | 6.1 | 10.7 | 58.3 | 85.7 |
| | | | ✓ | 154.4 | 6.6 | 11.5 | 57.9 | 85.4 |

(b) STSS Combinations

| $n$=1 | 2 | 3 | 4 | FLOPs | TP | Mem | SSV1 | D48 |
|---|---|---|---|---|---|---|---|---|
| ✓ | ✓ | | | 151.5 | 5.6 | 9.9 | **60.0** | **87.7** |
| ✓ | | ✓ | | 152.9 | 6.1 | 10.7 | 59.4 | 87.2 |
| ✓ | | | ✓ | 154.4 | 6.6 | 11.5 | 59.1 | 86.6 |
| | ✓ | ✓ | | 152.9 | 6.1 | 10.7 | 58.6 | 86.2 |
| ✓ | ✓ | ✓ | | 152.9 | 6.1 | 10.7 | 59.3 | 87.6 |

(c) High-Order STSS Transformation

| fusion | $\mathbf{S}^{(2)}$ | SSV1 | D48 |
|---|---|---|---|
| 1st STSS only | - | 59.0 | 86.3 |
| MLP | $f(\texttt{MLP}([\mathbf{F}, \mathbf{M}^{(1)}]))$ | 59.1 | 86.6 |
| conv | $f(\texttt{Conv2d}([\mathbf{F}, \mathbf{M}^{(1)}]))$ | 59.2 | 86.6 |
| addition | $f(\mathbf{F} + \mathbf{M}^{(1)})$ | 59.4 | 86.8 |
| no-fusion | $f(\mathbf{M}^{(1)})$ | **60.0** | **87.7** |

(d) Comparison to Other STSS Learning Methods

| method | FLOPs | TP | Mem | SSV1 | D48 |
|---|---|---|---|---|---|
| 1st STSS only | 150.0 | 5.1 | 9.0 | 59.0 | 86.3 |
| + R(2+1)D | 151.1 | 5.8 | 9.7 | 59.1 | 86.7 |
| + Fact Attn. | 151.2 | 5.8 | 11.0 | 59.2 | 86.4 |
| + Diff. | 151.5 | 5.6 | 9.7 | 59.2 | 86.5 |
| + 2nd STSS | 151.5 | 5.6 | 9.9 | **60.0** | **87.7** |

Furthermore, we also validate consistent effectiveness of our method on other video tasks such as temporal action detection and generic event boundary detection. Please refer to Sec. C for the detail.

## 4.3 EFFICIENCY COMPARISON

We compare the efficiency of our MOSS models to existing efficient tuning methods (Lin et al., 2022; Pan et al., 2022; Qing et al., 2023; Yang et al., 2023; Yao et al., 2023) in terms of FLOPs, the number of trainable parameters, memory footprint, and accuracy on Something-Something V2. The results are summarized in Tab. 3. Compared to the baseline Side4Video (Yao et al., 2023), our MOSS-S model significantly reduces the number of trainable parameters and memory footprints by 71% and 48% respectively, while achieving better performance. Across all efficiency metrics, MOSS-S shows the best trade-off among the compared methods. Scaling up to MOSS-B, we obtain superior performance with a competitive efficiency. Similar trends are observed for the ViT-L scale models as well. Our MOSS-M model reduces the number of parameters and memory footprints by 76% and 52% respectively, while maintaining competitive performance. While ST-Adapter requires fewer parameters and less FLOPs than MOSS-M, it has $2.9\times$ larger memory consumption. By scaling up to MOSS-L, we can widen the accuracy gap with a favorable efficiency. These results demonstrate the effectiveness of our lightweight yet high-performing MOSS modules in significantly boosting efficiency across different model scales.

## 4.4 ANALYSIS OF HIGH-ORDER STSS

We here provide in-depth analyses of high-order STSS on Something-Something V1 and Diving48 using MOSS-S with 8 and 32 frames, respectively.

**Effect of High-Order STSS.** In Tab. 4a, we show the individual effects of incorporating STSS at different orders. Compared to the baseline without any STSS, the 1st-order STSS significantly improves the top-1 accuracy by 2.1%p on Something-Something V1, highlighting the effectiveness of the motion information it captures. The 2nd- & 3rd-order STSS also demonstrate distinct improvements in accuracy by 1.8%p and 1.4%p, respectively. This indicates that high-order STSSs provide valuable cues for effective temporal understanding. We also observe that the 4th-order STSS is still beneficial, obtaining a 1.0%p gain, which is relatively smaller compared to the lower orders. Diving48 exhibits a similar trend, confirming the consistent benefits of higher-order STSS in temporal modeling.

**Mixed-Order STSS.** In Tab. 4b, we investigate the effect of combining different STSS orders to evaluate their complementarity. Fixing the 1st-order STSS, we add higher-order STSS features one by one. The results show that incorporating the 2nd and 3rd orders alongside the 1st order leads to further performance enhancements, indicating that they provide complementary temporal dynamics to the basic motion. In contrast, the 4th-order STSS does not yield meaningful improvements. We also observe that combining the 2nd and 3rd orders without the 1st-order STSS does not lead to additional performance gain compared to using either order individually. This lack of complementarity

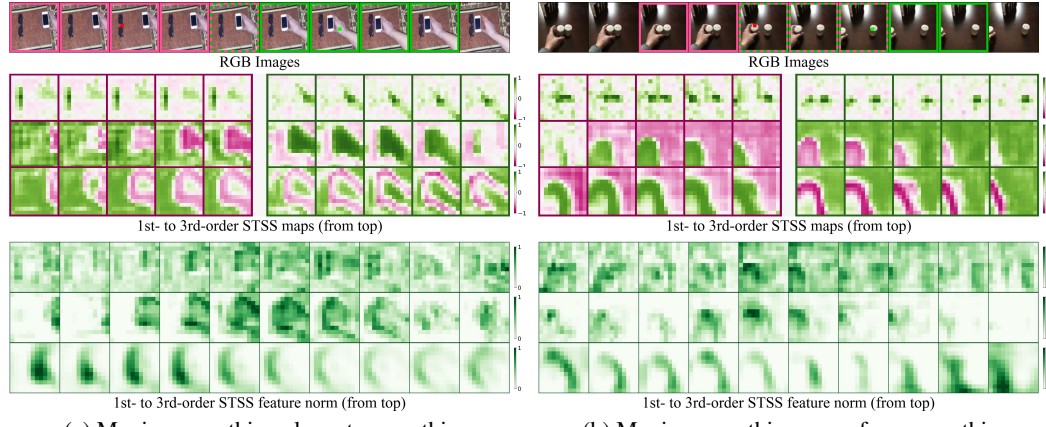

(a) Moving something closer to something        (b) Moving something away from something

Figure 5: **STSS visualization**. RGB frames at the top where two queries and their spatio-temporal matching regions are marked in red and green respectively. The subsequent rows show STSS maps for the two queries and L2-norm of feature maps across 1st-, 2nd-, and 3rd-order. Best viewed in pdf.

may indicate that these two orders capture redundant temporal dynamics despite their conceptual hierarchical differences. Consequently, combining STSS features from the 1st to 3rd orders together results in suboptimal performance. Based on the above, we use the combination of 1st-order and 2nd-order STSS features by default.

**High-Order STSS Transformation.** We compare transformation methods for computing the 2nd-order STSS $S^{(2)}$: 'MLP', 'conv', 'addition', and 'no-fusion' (Eq. 3). The first three methods combine the original visual feature $F$ with the 1st-order STSS feature $M^{(1)}$ before calculating $S^{(2)}$, while the last computes $M^{(2)}$ purely from $M^{(1)}$ without visual input. Intuitively, one might expect that augmenting the 1st-order STSS features with visual features would provide richer context for the 2nd-order STSS, leading to better performance. However, Tab. 4c shows the opposite: the 'no-fusion' significantly outperforms others. This suggests that the 1st- and 2nd-order STSS features capture complementary dynamics. Thus, maintaining the distinctness of multi-order STSS features—without merging them with the visual features—is key to fully exploiting their unique motion cues.

**Comparison to Other STSS Learning Methods.** In Tab. 4d, we delve into the effectiveness of high-order STSS by comparing different STSS learning methods. We keep the 1st STSS encoder fixed and replace the 2nd STSS encoder with R(2+1)D convolution (Tran et al., 2018), factorized self-attention (Arnab et al., 2021), and frame-wise difference calculation (Wu et al., 2023). Tab. 10c shows that other STSS learning methods provide marginal gains while our 2nd-order STSS encoder significantly boosts accuracy. This indicates that the 2nd-order STSS encoder captures complementary temporal dynamics distinct to the 1st-order.

**Visualization of High-Order STSS.** In Fig. 5, we present visualization results of 1st- to 3rd-order STSSs on Something-Something V1 to analyze their distinct contributions on temporal understanding. We observe that the 1st-order STSS identifies basic motions of objects (2nd row, Fig. 5a), but struggles to distinguish between visually similar objects with different motion patterns (2nd row, Fig. 5b). In contrast, the 2nd-order STSS overcomes this limitation by segmenting regions based on their motion patterns, effectively distinguishing visually similar objects (3rd row, Fig. 5b) and background regions (3rd row in Fig. 5a). The 3rd-order STSS further groups regions with similar motion segments, revealing motion boundaries where dis-occluded regions exhibit distinct motion segment patterns (4th row). We also visualize the L2-norm of STSS feature maps across different orders and then observe that higher-order STSSs, especially the 2nd-order, effectively suppress static regions while highlighting moving objects and their motion boundaries (5th & 6th rows), maintaining robustness under background clutter. These complementary contributions of high-order STSS beyond basic motion enable a comprehensive video understanding.

We present additional ablation studies in Secs. C.3 & D, more qualitative results in Sec. E, and in-depth discussions in Sec. F in our Appendix.

Table 5: **Results on FAVOR-Bench and MotionBench-Dev.** FAVOR-Bench consists of six tasks: Action Sequence (AS), Camera Motion (CM), Holistic Action Classification (HAC), Multiple Action Details (MAD), Non-subject Motion (NSM), and Single Action Detail (SAD). MotionBench-Dev includes six tasks as well: Motion Recognition (MR), Location-related Motion (LM), Camera Motion (CM), Motion-related Objects (MO), Action Order (AO), and Repetition Count (RC).

| method | FAVOR-Bench | | | | | | | MotionBench-Dev | | | | | |
|---|---|---|---|---|---|---|---|---|---|---|---|---|---|
| | all | AS | CM | HAC | MAD | NSM | SAD | all | MR | LM | CM | MO | AO | RC |
| VideoLLaMA3-2B | 42.2 | 43.8 | 27.3 | 44.4 | 48.1 | 45.3 | 42.8 | 50.2 | 54.9 | 54.2 | 36.1 | 68.3 | 37.0 | 27.3 |
| + FAVOR-Train | 45.5 | 44.8 | 27.8 | 54.5 | 51.3 | **54.7** | 45.3 | 51.4 | 55.8 | 54.4 | 36.4 | **68.8** | 38.3 | 33.0 |
| + FAVOR-Train + MOSS | **46.6** | **46.8** | **28.8** | **55.0** | **52.2** | 53.3 | **45.7** | **54.2** | **59.7** | **55.7** | **48.3** | 68.1 | **38.5** | **34.0** |

### 4.5 HIGH-ORDER STSS IN VIDEO LLMS

While Video LLMs demonstrate strong capability in story- or event-level temporal reasoning over long videos, recent studies (Tu et al., 2025; Hong et al., 2025) show they still struggle with fine-grained motion understanding. We here investigate whether integrating MOSS into Video LLMs can enhance fine-grained motion-level reasoning by providing primitive temporal cues to the LLM.

**Datasets.** *FAVOR-Bench* (Tu et al., 2025) evaluates fine-grained motion-level reasoning in Video LLMs, comprising 1,776 videos and 8,184 multiple-choice QA pairs across 6 motion-related tasks. We use 15K samples from the publicly released training set, *FAVOR-Train*, for fine-tuning. *Motion-Bench* (Hong et al., 2025) is an another recent benchmark designed for measuring fine-grained motion understanding, consisting of 5,385 videos and 8,052 QA pairs across 6 tasks. We evaluate our model on the dev split containing 4,018 questions.

**Implementation Details.** We adopt VideoLLaMA3-2B (Zhang et al., 2025) as our baseline. A single MOSS module is inserted after the 6th layer of the SigLip vision encoder (Zhai et al., 2023) to extract multi-order STSS features. The resulting features are added before the projector to inject early motion cues into the LLM. We set $(L, U, V) = (7, 11, 11)$ with feature dimension $D = 256$, and initialize the final FC layers to zeros to stabilize early training. The model is fine-tuned using LoRA with learning rates of 1e-3, 1e-4, and 1e-5 for MOSS, projector, and LLM LoRA weights, respectively, over 1,000 iterations. For both training and testing, all frames are resized such that the shorter side is 224 and sampled at 2 FPS.

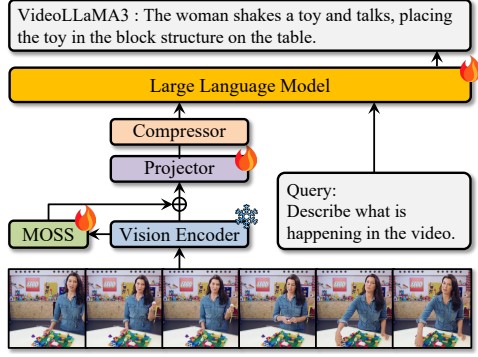

Figure 6: **VideoLLaMA3 with MOSS.** MOSS is integrated with the vision encoder and provides early motion cues for advanced temporal reasoning in LLM.

**Results.** We summarize the results in Tab. 5. Compared to VideoLLaMA3-2B fine-tuned on the same data, incorporating MOSS improves overall accuracy by 1.1%p on FAVOR-Bench. In addition, we directly evaluate on MotionBench without fine-tuning to validate generalizability. MOSS improves overall accuracy by 2.8%p, with notable gains on tasks requiring complex motion-level reasoning, including motion recognition, location-related motion, camera motion, and repetition counting, demonstrating that MOSS enhances motion understanding in a generalizable manner. Importantly, these improvements are achieved with 13.7M additional parameters (0.6% of the total parameters) and 8 GPU-hours only for training. This makes MOSS substantially more efficient than existing approaches (Liu et al., 2024b; Nie et al., 2024; Rasekh et al., 2025) that require large-scale re-training with massive video datasets, while offering strong generalization to unseen benchmarks.

## 5 CONCLUSION

We have provided an in-depth analysis of high-order space-time self-similarities and demonstrated that each order captures unique and complementary aspects of temporal dynamics. We introduced MOSS, a lightweight neural module that learns and integrates multi-order STSS features as multi-faceted motion representations. By incorporating MOSS into a ladder side tuning framework, we achieved strong performance on various action recognition benchmarks, significantly improving the memory-accuracy trade-off. These results highlight the potential of high-order STSS in capturing complex motion patterns, demonstrating its role in comprehensive video understanding.

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

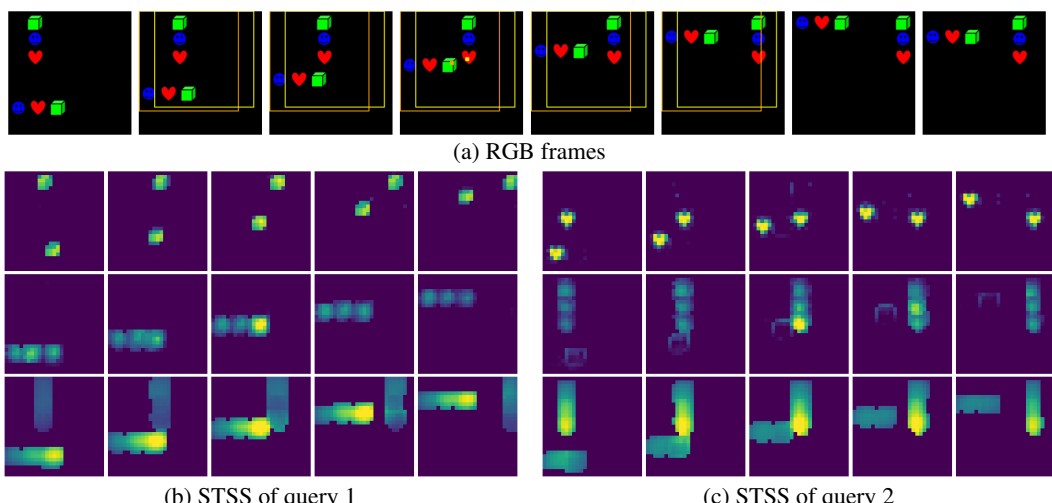

(a) RGB frames

(b) STSS of query 1                    (c) STSS of query 2

Figure 7: **Visualization of STSS tensors.** (a) Input RGB frames, where two different queries and their spatio-temporal matching regions. (b) 1st- to 3rd-order STSS maps of the brown query. (c) 1st- to 3rd-order STSS maps of the yellow query.

## A    ILLUSTRATION OF HIGH-ORDER STSS

We present a toy example with a simplified video clip to clarify the characteristics of high-order STSS in modeling temporal dynamics, as described in Secs. 3.1 and 3.2.

**Experimental Setup.** We synthesize a controlled video clip with two sets of moving objects, each comprising three objects: a blue circle, a red heart, and a green cube (Fig. 7a). Objects within each set share the same motion—either horizontal or vertical movement. We select two green cubes as queries and visualize their STSS maps from the 1st- to the 3rd-order (Figs. 7b and 7c). Here, we define the STSS encoding function $g$ as vectorization over $(L, U, V)$ dimensions, *i.e.*, $g : \mathbb{R}^{T \times H \times W \times L \times U \times V} \rightarrow \mathbb{R}^{T \times H \times W \times (LUV)}$.

**Characteristics of STSS at Different Orders.** The 1st-order STSS captures the motion flow of the query by establishing correspondences based on appearance across frames. However, it struggles to distinguish between visually similar objects in different sets, capturing the motion of other unintended objects. The 2nd-order STSS addresses this limitation by computing similarities based on motion patterns rather than appearance. It effectively identifies set of objects that shares similar motion with the query, distinguishing visually similar objects moving differently. The 3rd-order STSS extends this by grouping regions based on these motion segments, highlighting overall motion patterns across all object sets and enabling a higher-level understanding of motion. This progression—from capturing motion flows to identifying motion segments to understanding motion at the object set level—reveals diverse aspects of temporal dynamics.

## B    IMPLEMENTATION DETAILS

In Tables 6 and 7, we provide detailed model configurations and training hyperparameters across different model scales and datasets. All models are trained using 8 NVIDIA RTX 6000 Ada GPUs.

**License.** We implement our model based on Side4Video (Yao et al., 2023) in PyTorch[1] under MIT license. The datasets used in our experiments are publicly available for academic research. Kinetics-400 is available under CC BY 4.0 license, Something-Something uses a custom academic license[2], Diving48's license is unknown, and FineGym is under CC BY-NC 4.0 license.

---

[1] https://github.com/HJYao00/Side4Video
[2] https://www.qualcomm.com/content/dam/qcomm-martech/dm-assets/documents/jester_something_something_exercise_research_license_final_qti_28jul2022.pdf

Table 6: **Model configurations**. "TP" indicates the number of trainable parameters. FLOPs are measured using 8 frames.

| methods | image encoder | | | temporal encoder | | | FLOPs (G) | TP (M) |
|---|---|---|---|---|---|---|---|---|
| | layer | dim | head | layer | dim | head | | |
| MOSS-S | 12 | 768 | 12 | 6 | 192 | 3 | 151 | 6 |
| MOSS-B | 12 | 768 | 12 | 12 | 320 | 5 | 179 | 22 |
| MOSS-M | 24 | 1024 | 16 | 12 | 320 | 5 | 707 | 24 |
| MOSS-L | 24 | 1024 | 16 | 24 | 448 | 7 | 833 | 82 |

Table 7: **Training configurations on Kinetics-400, Something-Something V1&V2, Diving48, and FineGym**.

| Setting | Kinetics-400 | | Something V1 & V2 | | | | Diving48 | | FineGym | |
|---|---|---|---|---|---|---|---|---|---|---|
| | B | L | S | B | M | L | B | L | B | L |
| *Optimization* | | | | | | | | | | |
| batch size | 128 | 96 | 128 | | | 96 | 128 | 80 | 128 | 80 |
| epochs | 30 | | 40 | | | 30 | 30 | | 30 | |
| learning rate | 1e-3 | 2e-4 | 1e-3 | | | 2e-4 | 1e-3 | 2e-4 | 1e-3 | 2e-4 |
| lr schedule | cosine decay | | | | | | | | | |
| optimizer | AdamW ($\beta_1 = 0.9$, $\beta_2 = 0.999$) | | | | | | | | | |
| weight decay | 0.15 | | | | | | | | | |
| warmup epochs | 4 | | | | | | | | | |
| pre-train | CLIP400M | | CLIP400M | | | | CLIP400M+K400 | | CLIP400M+K400 | |
| *Augmentation* | | | | | | | | | | |
| sampling | dense (stride=8) | | uniform | | | | uniform | | uniform | |
| resize | RandomResizedCrop | | | | | | | | | |
| RandAugment | - | | rand-m7-n4-mstd0.5-inc1 | | | | - | | - | |
| random flip | 0.5 | | | | | | | | | |
| label smoothing | 0.1 | | | | | | | | | |
| repeated Aug. | 2 | | 2 | | | | 1 | | 1 | |
| gray scale | 0.2 | | - | | | | - | | - | |
| *MOSS module* | | | | | | | | | | |
| window $(L, U, V)$ | (5, 9, 9) | | | | | | | | | |
| # channels $D$ | 64 | 96 | 64 | | | 96 | 64 | 96 | 64 | 96 |
| # Enc. Blocks | 3 | | | | | | | | | |
| position $k$ | 4 | 8 | 4 | | | 8 | 4 | 8 | 4 | 8 |

**Source Code.** Code and logs are provided in our supplementary material.

## C  ADDITIONAL EXPERIMENTS

This section presents additional experimental results, including temporal action detection, generic event boundary detection, and ablation studies.

### C.1  TEMPORAL ACTION DETECTION

**Datasets.** *THUMOS-14* (Idrees et al., 2017) is a widely used benchmark for temporal action detection (TAD), containing 200 and 213 untrimmed videos for training and testing, annotated with 20 action classes.

**Implementation Details.** We adopt AdaTAD (Liu et al., 2024c) as our baseline, which adapts VideoMAE to a TAD model with *TIA adapters*. We insert a single MOSS module between the 6th VideoMAE block and the TIA adapter. Following Liu et al. (2024c), we train only the MOSS module and TIA adapters. We use 768 input frames with a resolution of 224×224 for training. For evaluation, we report the mean Average Precision (mAP) at different temporal Intersection over Union (tIoU) thresholds.

Table 8: **Results of temporal action detection on THUMOS-14**. We report mAP with tIoU thresholds from 0.3 to 0.7 and their average. * reproduced by out setup.

| method | backbone | THUMOS-14 | | | | | |
| | | 0.3 | 0.4 | 0.5 | 0.6 | 0.7 | Avg. |
|---|---|---|---|---|---|---|---|
| TALLFormer (Cheng & Bertasius, 2022) | VSwin-B | 76.0 | - | 63.2 | - | 34.5 | 59.2 |
| ActionFormer (Yang et al., 2024) | I3D | 82.1 | 77.8 | 71.0 | 59.4 | 43.9 | 66.8 |
| TriDet (Shi et al., 2023) | I3D | 83.6 | 80.1 | 72.9 | 62.4 | 47.4 | 69.3 |
| AdaTAD (Liu et al., 2024c) | VMAE-B | 87.0 | 82.4 | 75.3 | 63.8 | 49.2 | 71.5 |
| AdaTAD (Liu et al., 2024c) | VMAE-L | 87.7 | 84.1 | 76.7 | 66.4 | 52.4 | 73.5 |
| AdaTAD* (Liu et al., 2024c) | VMAE-B | 85.7 | 82.3 | 75.3 | 63.4 | 50.1 | 71.4 |
| +MOSS | VMAE-B | 87.3 | 82.9 | 75.7 | 64.8 | 50.3 | **72.2** |
| AdaTAD* (Liu et al., 2024c) | VMAE-L | 87.5 | 83.7 | **77.6** | 66.5 | 51.9 | 73.5 |
| +MOSS | VMAE-L | **88.1** | **84.5** | **77.6** | **67.7** | **52.8** | **74.1** |

Table 9: **Results of GEBD on Kinetics-GEBD and TAPOS**. * reproduced by our setup. "Avg. F1" is an averaged F1 score with relative distance thresholds from 0.05 to 0.5. with 0.05 interval.

| method | Kinetics-GEBD | | TAPOS | |
| | F1@0.05 | Avg. F1 | F1@0.05 | Avg. F1 |
|---|---|---|---|---|
| Temporal Perceiver (Tan et al., 2023) | 74.8 | 86.0 | 55.2 | 73.2 |
| DDM-Net (Tang et al., 2022) | 76.4 | 87.3 | 60.4 | 72.8 |
| SC-Transformer (Li et al., 2022a) | 77.7 | 88.1 | 61.8 | 74.2 |
| BasicGEBD (Zheng et al., 2024) | 76.8 | 86.6 | 60.0 | 71.0 |
| EfficientGEBD (Zheng et al., 2024) | 78.3 | 88.3 | 63.1 | 74.8 |
| BasicGEBD* (Zheng et al., 2024) | 76.9 | 86.4 | 60.1 | 71.2 |
| BasicGEBD+MOSS | **79.4** | **89.0** | **68.5** | **76.6** |

**License.** We implement MOSS on OpenTAD (Liu et al., 2025) repository[3] which is available under the Apache 2.0 license. While the THUMOS-14 dataset's license is unknown, it is widely used in research for temporal action detection benchmarking.

**Results.** Table 8 presents the TAD results on THUMOS-14. Our MOSS-enhanced model consistently outperforms AdaTAD baseline across all tIoU thresholds, reaching 72.2% and 74.1% average mAP with VideoMAE-B and L backbones respectively. These results demonstrate the versatility of MOSS module on longer video understanding.

## C.2 GENERIC EVENT BOUNDARY DETECTION

Generic event boundary detection (GEBD) aims to localize event boundaries in a video, such as changes in subject appearance or motion patterns, segmenting it into distinct and meaningful chunks. Accurate detecting these instantaneous changes is crucial for effective event segmentation.

**Datasets.** *Kinetics-GEBD* (Shou et al., 2021) is the largest GEBD dataset, consisting of 55K videos with 1.3M taxonomy-free event boundaries including action and object changes. *TAPOS* (Shao et al., 2020b) comprises 15K Olympic sports videos with 21 distinct action classes. Following Shou et al. (2021), we re-design TAPOS for GEBD task by trimming each action instance.

**Implementation Details.** We employ BasicGEBD-L4 (Zheng et al., 2024) as backbone and add a single MOSS module after the 2nd stage of ResNet-50 and train the entire network end-to-end following the training protocols in Zheng et al. (2024). For evaluation, we measure F1 score with relative distance 0.05 and average F1 score from 0.05 to 0.5 with 0.05 interval.

**License.** We implement MOSS on EfficientGEBD (Zheng et al., 2024) repository[4]. Both Kinetics-GEBD and TAPOS datasets are available under CC BY-NC 4.0 license.

**Results.** Table 9 summarizes the results on Kinetics-GEBD and TAPOS. Compared to our baseline (BasicGEBD), our MOSS module substantially improves performance at F1@0.05 scores increasing by 2.5%p and 8.4%p on Kinetics-GEBD and TAPOS, respectively, achieving new state-of-the-art

---

[3] https://github.com/sming256/OpenTAD
[4] https://github.com/StanLei52/EfficientGEBD

Table 10: **Additional ablation studies on Something-Something V1 and Diving48.** All experiments are conducted with MOSS-S taking 8 and 32 frames as input, respectively. "FLOPs", "TP", and "Mem" respectively indicate FLOPs (G), trainable parameters (M), and memory footprint (GB) using 8 frames. Memory footprint is measured using a batch size of 32 for a single GPU machine.

(a) Temporal window size $L$

| 1st | 2nd | FLOPs | TP | Mem | SSV1 | D48 |
|---|---|---|---|---|---|---|
| 3 | 5 | 150.9 | 5.4 | 9.8 | 59.5 | 87.0 |
| 5 | 5 | 151.5 | 5.6 | 9.9 | **60.0** | 87.7 |
| 7 | 5 | 152.0 | 5.7 | 10.5 | 59.9 | 87.8 |
| 5 | 3 | 150.9 | 5.4 | 9.8 | 59.2 | 86.5 |
| 5 | 7 | 152.0 | 5.7 | 10.5 | 59.7 | **88.0** |

(b) Module positions

| pos | FLOPs | TP | Mem | SSV1 | D48 |
|---|---|---|---|---|---|
| 2 | 151.5 | 5.6 | 9.9 | 59.1 | 85.4 |
| 4 | 151.5 | 5.6 | 9.9 | 60.0 | 87.7 |
| 6 | 151.5 | 5.6 | 9.9 | 58.9 | 86.5 |
| 8 | 151.5 | 5.6 | 9.9 | 58.2 | 86.4 |
| 4,6 | 154.5 | 6.7 | 11.9 | **60.1** | 88.1 |

(c) Comparison to Other Temporal Modules

| method | FLOPs | TP | Mem | SSV1 | D48 |
|---|---|---|---|---|---|
| baseline | 148.4 | 4.5 | 8.0 | 56.9 | 85.0 |
| R(2+1)D | 151.4 | 6.4 | 9.6 | 57.5 | 86.1 |
| Fact Attn. | 151.8 | 6.6 | 11.5 | 57.4 | 85.9 |
| Local Attn. | 150.6 | 5.6 | 11.0 | 57.5 | 85.5 |
| SELFY | 151.1 | 5.1 | 11.2 | 59.2 | 87.0 |
| ATM | 153.2 | 6.0 | 12.1 | 59.6 | 87.2 |
| MOSS (ours) | 151.5 | 5.6 | 9.9 | **60.0** | **87.7** |

(d) Finetuning Methods.

| FT | method | FLOPs | TP | Mem | SSV1 | D48 |
|---|---|---|---|---|---|---|
| full FT | ViT-B | 140.7 | 86.4 | 28.3 | 51.9 | 84.2 |
| | + MOSS | 144.1 | 87.6 | 30.3 | **59.6** | **87.8** |
| PEFT | AIM ViT-B | 207.9 | 14.3 | 38.6 | 54.8 | 87.3 |
| | + MOSS | 211.3 | 15.6 | 40.9 | 57.4 | **89.4** |
| LST | Side4Video | 148.4 | 4.5 | 8.0 | 56.9 | 85.0 |
| | + MOSS | 151.5 | 5.6 | 9.9 | **60.0** | **87.7** |
| | DiST | 163.1 | 19.0 | 11.1 | 55.6 | 86.3 |
| | + MOSS | 165.5 | 20.3 | 12.8 | **58.5** | **88.9** |

(e) Different image encoders

| ViT-B | FLOPs | TP | Mem | SSV1 | D48 |
|---|---|---|---|---|---|
| MAE | 148.4 | 4.5 | 8.0 | 53.1 | 83.6 |
| +1st STSS | 150.0 | 5.1 | 9.0 | 54.9 | 86.3 |
| +MOSS (ours) | 151.5 | 5.6 | 9.9 | **56.0** | **87.2** |
| DINO | 148.4 | 4.5 | 8.0 | 53.5 | 84.1 |
| +1st STSS | 150.0 | 5.1 | 9.0 | 55.3 | 85.0 |
| +MOSS (ours) | 151.5 | 5.6 | 9.9 | **56.6** | **86.3** |
| CLIP | 148.4 | 4.5 | 8.0 | 56.9 | 85.0 |
| +1st STSS | 150.0 | 5.1 | 9.0 | 59.0 | 86.3 |
| +MOSS (ours) | 151.5 | 5.6 | 9.9 | **60.0** | **87.7** |

results. These significant improvements demonstrate that our proposed module effectively captures fine-grained temporal changes in both motion and objects, which is crucial for accurate event boundary detection.

## C.3 ADDITIONAL ABLATION EXPERIMENTS

We present additional ablation studies on Something-Something V1 and Diving48. Unless otherwise specified, we follow the experimental protocols in Secs. 4.1 and 4.4.

**Temporal Window Size $L$.** In Table 10a, we examine the effect of the size of temporal window $L$ for STSS transformation while keeping the spatial window size fixed as $(U, V) = (9, 9)$. We first vary the temporal window size of the 1st-order STSS keeping that of the 2nd-order STSS constant. We observe that increasing $L$ from 3 to 5 improves performance by capturing longer-range temporal dynamics. However, performance saturates when $L$ exceeds 5, providing no significant additional gains. Similarly, varying the temporal window size of the 2nd-order STSS while fixing that of the 1st-order yields comparable results. Based on these results, we set the temporal window size $L = 5$ for both the 1st- and 2nd-order STSS.

**Module Position.** In Table 10b, we examine the effect of different positions and the numbers of the MOSS module. The results show that the MOSS module is beneficial for all the cases but the performance depends on the position of the module. MOSS module inserted after the 4th image encoder block performs the best. We interpret these results as a trade-off between the robustness of the STSS transformation and the effectiveness of temporal modeling; Inserting the module too early may lead to noisy STSS transformation due to insufficient visual semantics in the feature maps, whereas inserting the module too late limits the capacity for temporal modeling because fewer temporal blocks remain to process the enriched features. Given marginal gain of using multiple modules, we add a single module after 4th block by default considering the efficiency.

**Comparison to Existing Temporal Modules.** We compare our method to other temporal modeling modules (Arnab et al., 2021; Kwon et al., 2021; Tran et al., 2019; 2018; Wu et al., 2023) with similar computational costs. we replace the MOSS module with different modules, including spatio-temporal convolution (Tran et al., 2018), factorized spatio-temporal attention (Arnab et al., 2021), local attention with a spatio-temporal window $(L, U, V)$, and the other STSS learning blocks (Kwon et al., 2021; Wu et al., 2023). SELFY and ATM extract 1st-order STSS features using convolutions, with ATM additionally performing frame-wise subtraction for richer dynamics. Table 10c shows that transforming STSS directly into motion features (Kwon et al., 2021; Wu et al., 2023) is more effective at capturing temporal dynamics than convolution or attention, consistent with prior work (Kwon et al., 2021). However, the effectiveness of SELFY (Kwon et al., 2021) and ATM (Wu et al., 2023)

is overshadowed by excessive memory overhead when applying a series of convolutions to large STSS tensor $\mathbf{S}$. In contrast, we use a simple FC layer to directly reduce the volume of $\mathbf{S}$. This enables memory-efficient processing of multi-order STSSs, leading to superior performance with less memory consumption.

**Finetuning Methods.** Although we integrate our MOSS module into LST framework (Yao et al., 2023) for efficient action recognition in previous experiments, MOSS is also compatible with various finetuning scenarios including full finetuning and parameter-efficient finetuning (PEFT) (Pan et al., 2022; Yang et al., 2023). Here we conduct experiments in such scenarios. For full finetuning, we add temporal convolution blocks and a single MOSS module to the CLIP-pretrained ViT-B (Radford et al., 2021) and train the entire network following Wu et al. (2023). For PEFT, we integrate MOSS into AIM (Yang et al., 2023) and train the module and adapters keeping the backbone frozen. For LST, we additionally conduct experiments on DiST (Qing et al., 2023) by inserting a single MOSS module between the spatial encoder and the integration branch. Table 10d shows that MOSS substantially improves performance with marginal computational overhead in all settings, demonstrating its flexibility in different image-to-video transfer methods.

**Spatial Encoders.** In Table 10e, we evaluate MOSS on ViT-B pretrained on three different objectives: CLIP (Radford et al., 2021), DINO (Caron et al., 2021) and MAE (He et al., 2022). The results show that both 1st- and 2nd-order STSS consistently improve performance across all pre-training objectives. Among the three encoders, CLIP achieves the best performance due to its generalizable visual representations, leading us to adopt it as our default spatial encoder.

# D    PER-CLASS ANALYSIS

We here provide a statistical analysis across diverse action classes, offering a comprehensive understanding of when higher-order STSS becomes effective. To this end, we measure the differences in accuracies of each *action groups* in Something-Something V1 (Goyal et al., 2017) when incorporating the 2nd-order STSS on top of the 1st-order STSS. The results are summarized in Fig. 8. Among the 50 action groups, we observe accuracy improvements in 39 groups and drops in 10 groups. Specifically, we find that the 2nd-order STSS is beneficial in understanding not only basic motions, *e.g.*, moving something or moving/touching a part of something (Fig. 9), but also more complex object-object interactions, *e.g.*, passing/hitting another object or moving two objects relative to each other (Fig. 10), (dis-)appearance events, *e.g.*, burying, covering, or dropping (Fig. 11a), and camera motion scenarios (Fig. 11b). These improvements indicate that the 2nd-order STSS captures complementary temporal dynamics beyond what the 1st-order STSS can capture. Meanwhile, the accuracy drops mainly in action groups such as squeezing, spinning, or twisting. This implies that when motion blur or severe deformation makes 1st-order STSS unreliable, motion segmentation in 2nd-order STSS becomes ambiguous resulting in limited benefits. This statitistical analysis reveals when higher-order STSS provides tangible benefits and when it becomes less effective, providing practical guidance for using higher-order STSS.

# E    ADDITIONAL VISUALIZATION RESULTS

We present additional visualization results on Something-Something V1 in Figs. 9-13. These figures visualize STSS maps and their feature L2 norms across the 1st to 3rd layers for videos including: simple motions (Fig. 9), object-object interactions (Fig. 10), sudden object appearance (Fig. 11a), camera motions (Fig. 11b), motion changes (Fig. 12), and background clutter (Fig. 13).

# F    LIMITATION AND FUTURE WORK

Our research explores the role of high-order STSS in learning video representations. While our theoretical analysis and the toy examples demonstrate that 3rd-order STSS has the potential to capture group-wise motion patterns, we observe that the learned model primarily utilizes 3rd-order STSS to capture motion boundaries for video action recognition (Figs. 5, 9-13). This limited utilization of 3rd-order STSS may explain why 2nd- and 3rd-order STSS are not complementary to each other (Table 4b) since 2nd-order STSS can already provides such boundary information implicitly. We conjecture that learning group-wise motion patterns may not provide significant benefits for

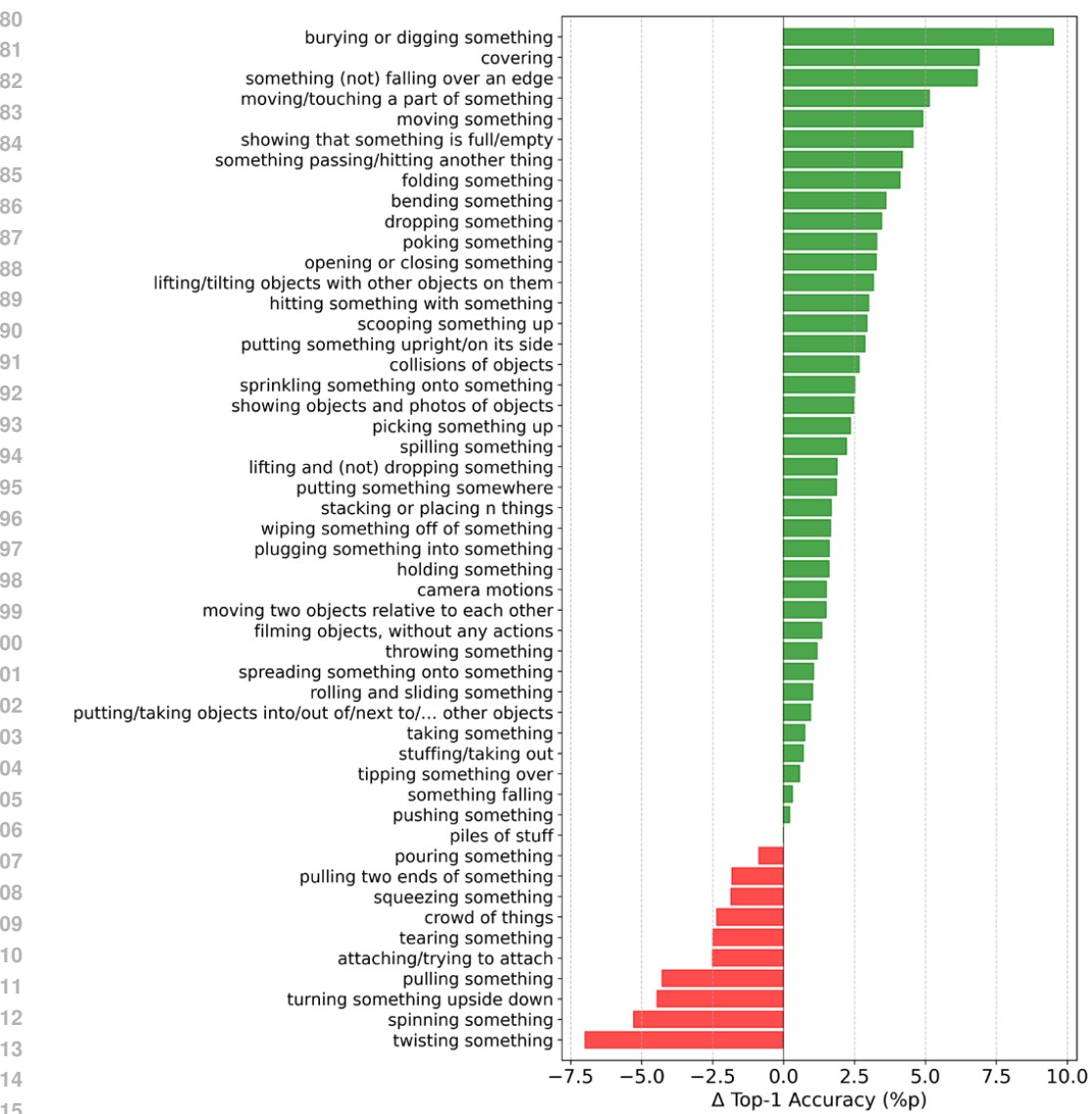

Figure 8: **Effects of 2nd-order STSS on Something-Something V1.** The figure shows the accuracy difference for each action group when incorporating the 2nd-order STSS on top of the 1st-order.

existing action recognition tasks. Future work should focus on developing new benchmarks where higher-order (3rd-order or beyond) temporal dynamics can demonstrate more meaningful benefits.

## G  REPRODUCIBILITY STATEMENT

We present detailed experimental setups including datasets, model configurations, and training hyperparameters in Secs. 4.1 and B. We also provide Pytorch implementation code and log files in our supplementary material. The code and checkpoints will be available publicly after acceptance.

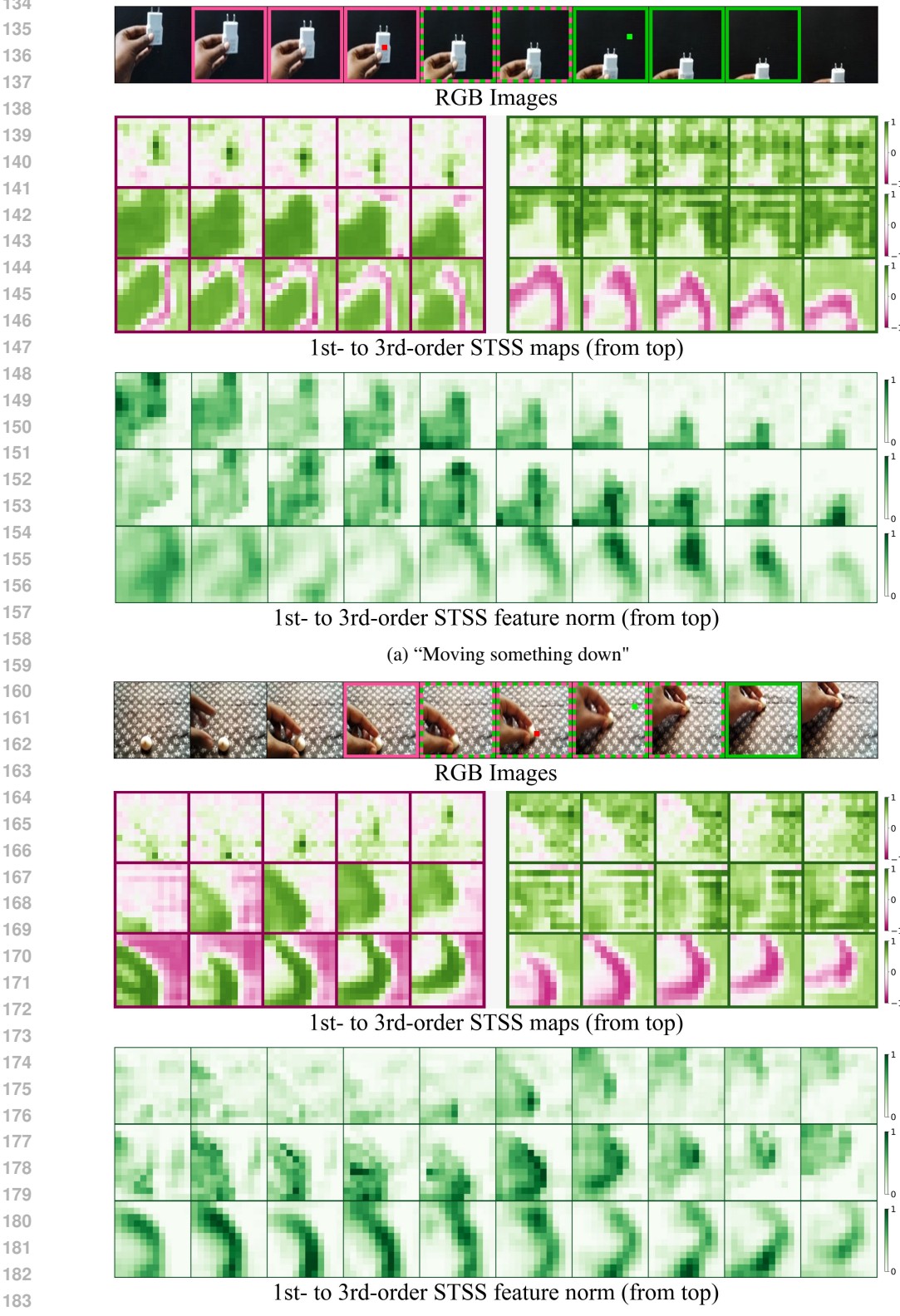

Figure 9: **STSS visualization**. RGB frames at the top show query locations and their spatio-temporal matching regions marked in red and green, respectively. The subsequent rows show STSS maps for the two queries and the L2-norm of feature maps from 1st- to 3rd-order STSSs. Best viewed in PDF.

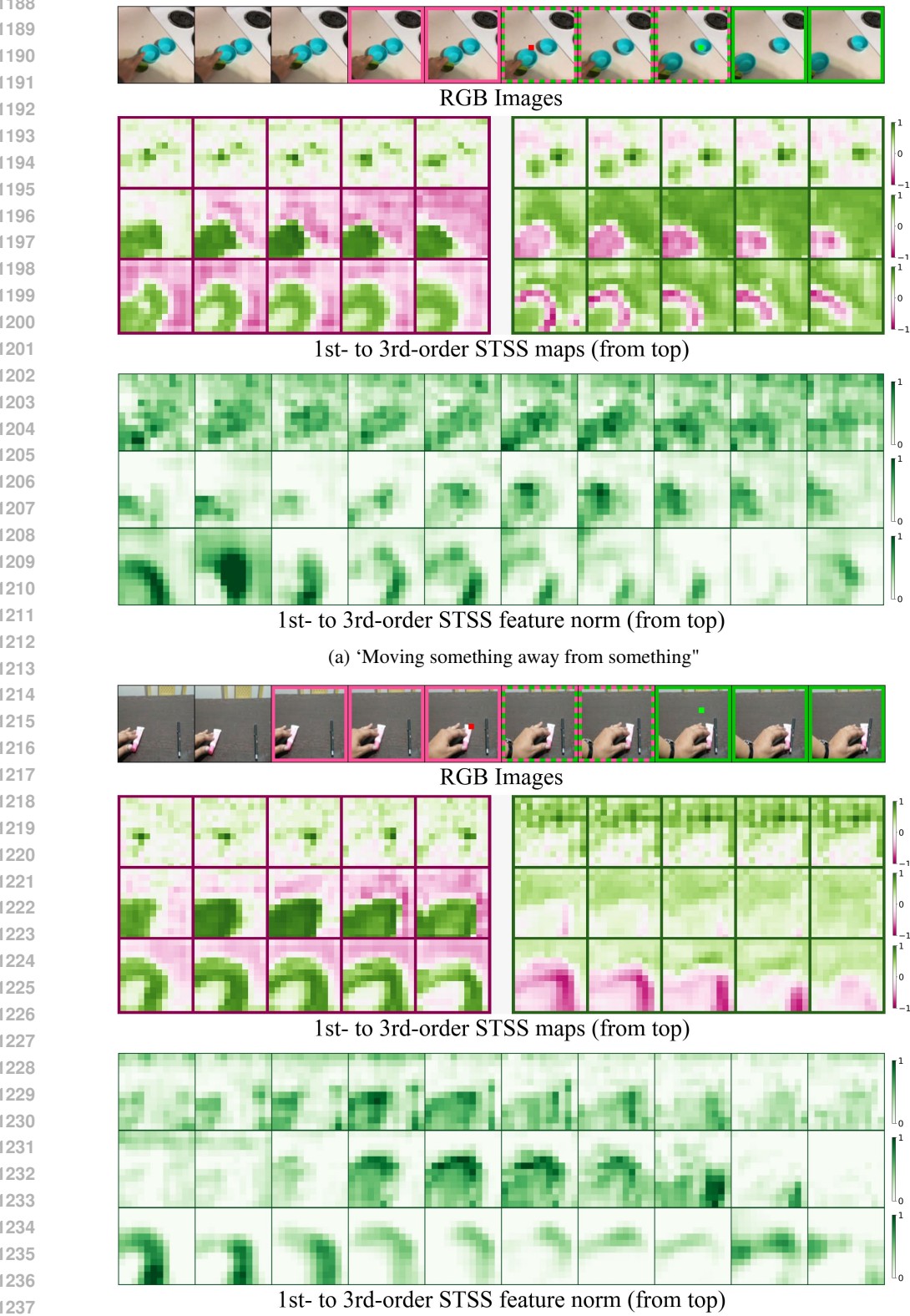

(a) 'Moving something away from something"

(b) "Moving something closer to something"

Figure 10: **STSS visualization**. RGB frames at the top show query locations and their spatio-temporal matching regions marked in red and green, respectively. The subsequent rows show STSS maps for the two queries and the L2-norm of feature maps from 1st- to 3rd-order STSSs. Best viewed in PDF.

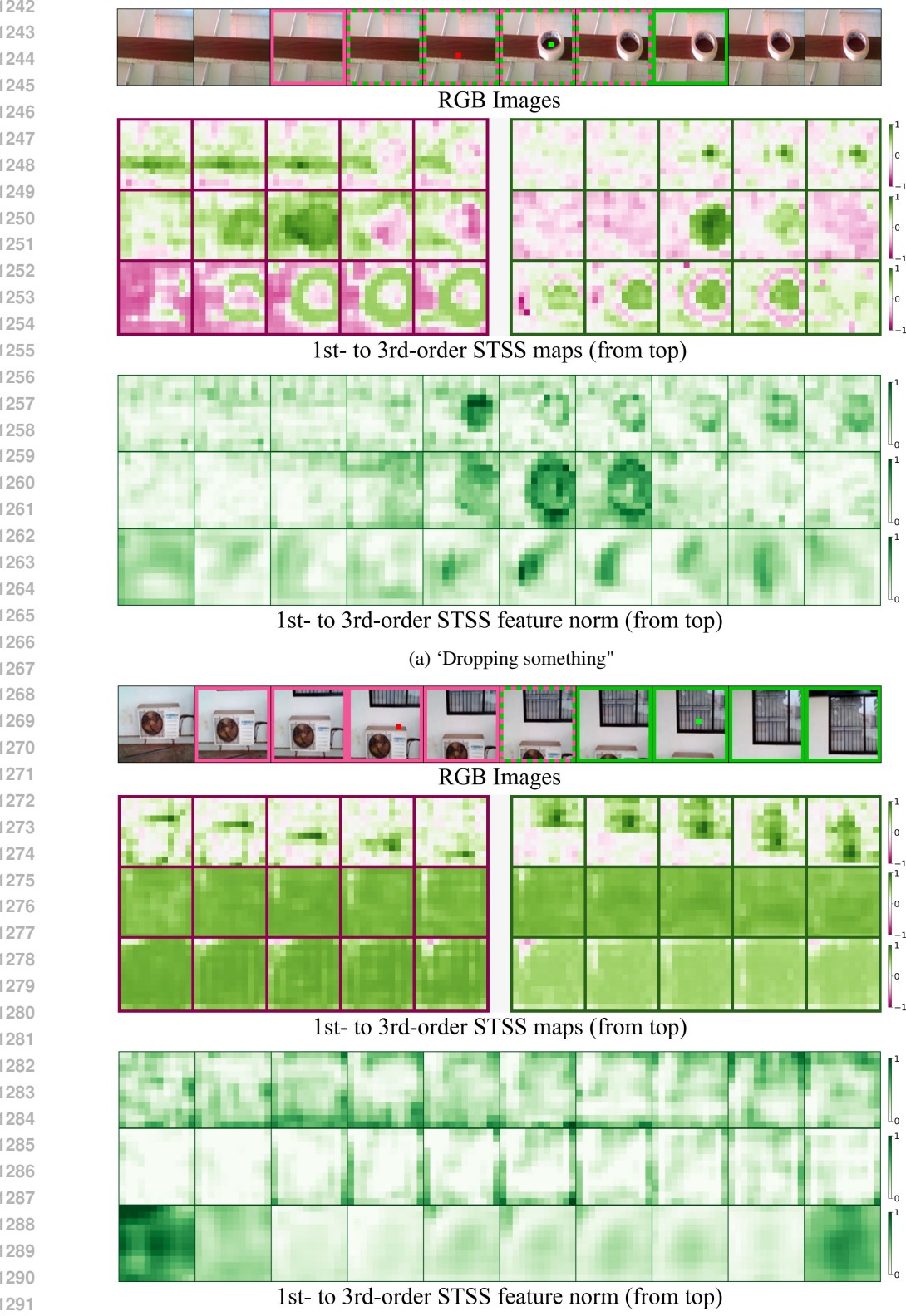

(a) 'Dropping something"

(b) "Turning the camera downwards while filming something"

Figure 11: **STSS visualization**. RGB frames at the top show query locations and their spatio-temporal matching regions marked in red and green, respectively. The subsequent rows show STSS maps for the two queries and the L2-norm of feature maps from 1st- to 3rd-order STSSs. Best viewed in PDF.

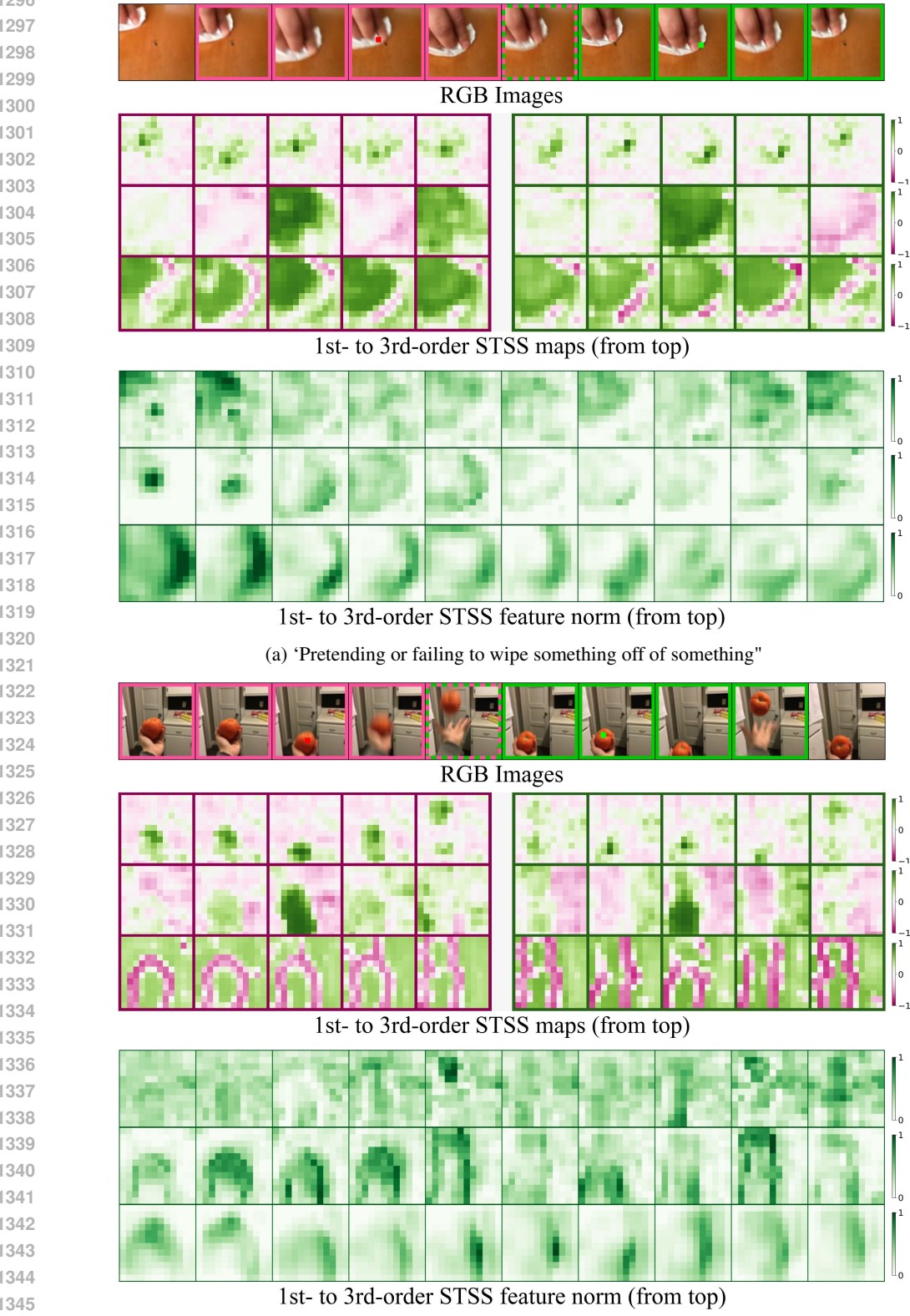

(a) 'Pretending or failing to wipe something off of something"

(b) "Throwing something in the air and catching it"

Figure 12: **STSS visualization**. RGB frames at the top show query locations and their spatio-temporal matching regions marked in red and green, respectively. The subsequent rows show STSS maps for the two queries and the L2-norm of feature maps from 1st- to 3rd-order STSSs. Best viewed in PDF.

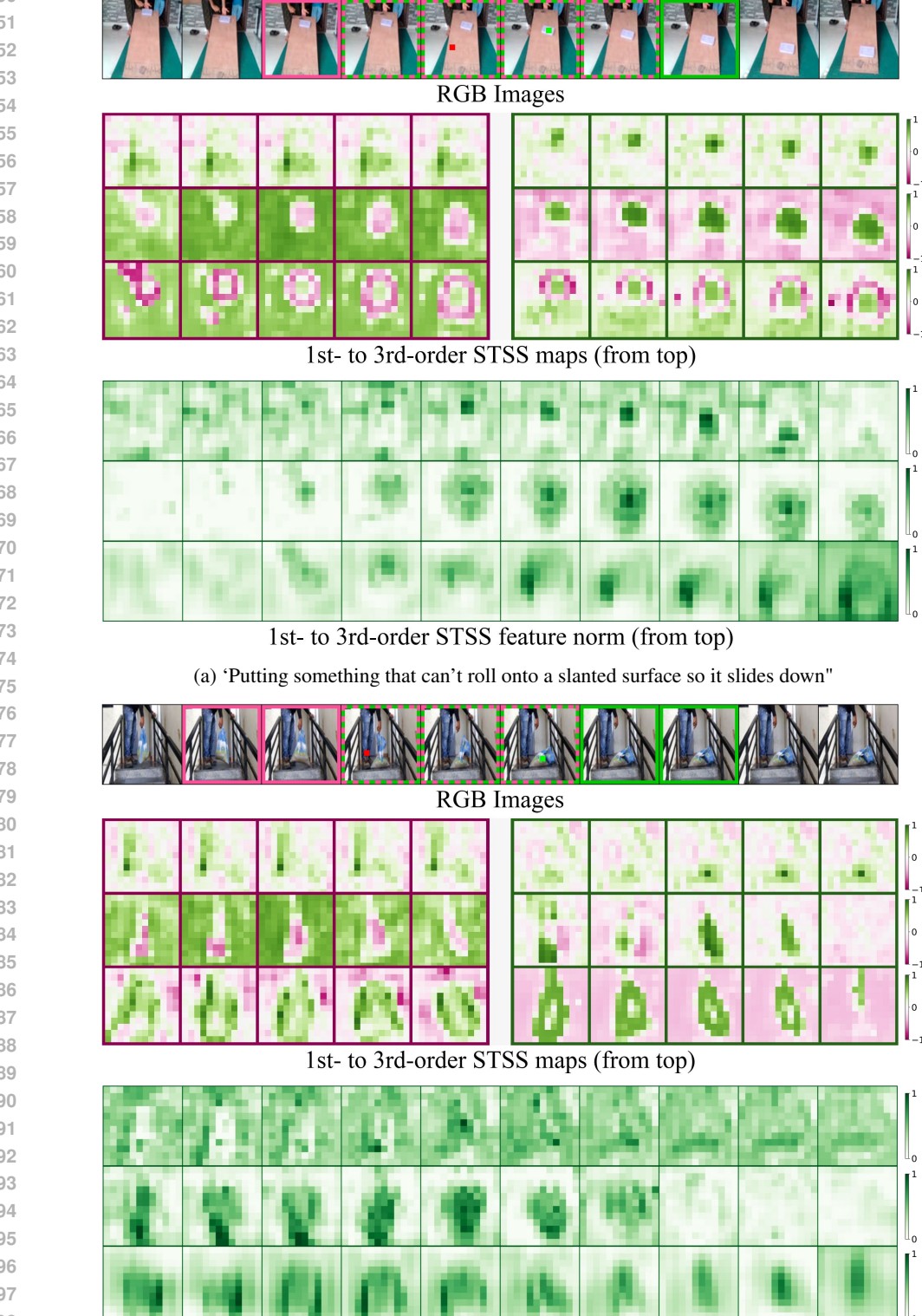

(a) 'Putting something that can't roll onto a slanted surface so it slides down"

(b) "Lifting something up completely then letting it drop down"

Figure 13: **STSS visualization**. RGB frames at the top show query locations and their spatio-temporal matching regions marked in red and green, respectively. The subsequent rows show STSS maps for the two queries and the L2-norm of feature maps from 1st- to 3rd-order STSSs. Best viewed in PDF.

