# OpenReview forum: "Exploring High-Order Self-Similarity for Video Understanding"
_ICLR.cc/2026/Conference — Submitted to ICLR 2026_

### Official Review · Reviewer_LQoE · 2025-10-20

**Soundness:** 3
**Presentation:** 3
**Contribution:** 2
**Rating:** 6
**Confidence:** 3

**Summary:**

This paper addresses conventional video understanding tasks by exploring high-order self-similarity. It begins by analyzing how high-order spatio-temporal self-similarities (STSS) facilitate learning temporal dynamics in videos, supported by vivid visualizations. Subsequently, it proposes MOSS, a lightweight neural module designed for comprehensive temporal understanding. Experimental results on widely-used benchmarks, including Kinetics-400, Something-Something V2 (SthSth-V2), Diving48, and FineGym, demonstrate the effectiveness and efficiency of the proposed approach.

**Strengths:**

1. This paper is well-written, presenting substantial qualitative and quantitative results that clearly support the effectiveness of the proposed high-order STSS for video understanding.

2. The chosen entry point, i.e., learning distinct high-order STSS representations particularly for dynamic temporal video understanding, is both novel and interesting. Furthermore, the visualizations of high-order STSS effectively illustrate their operational mechanisms.

3. This work is generally solid, as the proposed approach is thoroughly verified across multiple benchmarks and various video understanding tasks, including action recognition, temporal action detection, and generic event boundary detection.

**Weaknesses:**

1. My primary concern revolves around the recency of the compared methods. The main baselines, ATM and Side4Video, were both released in 2023, and generally, most works compared in this paper date from 2023 or earlier. This raises questions about the absence of comparisons with more recent advancements in the field. If there are indeed no significant video understanding (e.g., action recognition) works from 2024 or 2025 that are relevant, it would be important for the authors to clarify this. Otherwise, the lack of comparison with contemporary methods might suggest that incremental modifications to conventional and relatively smaller models may not be considered a substantial contribution to the broader video understanding area.

2. Regarding the overall architecture, could the authors explain the rationale behind bridging certain layers with MOSS modules while others are connected solely via Fully Connected (FC) layers? Is there any analysis presented to demonstrate that different layers exhibit varying preferences for motion-augmented visual features?

**Questions:**

Some additional typos:
1. Given the paper's analysis that the 2nd order STSS provides motion segment information, and the 3rd order STSS demonstrates the overall layouts of motion segments, it is puzzling why combining the 1st, 2nd, and 3rd order STSS does not lead to better results in Table 4(b). While the authors claim that "redundant temporal dynamics" are captured for the 2nd and 3rd orders, this explanation does not seem to align well with the vivid illustrations provided in the main paper.

2. How do the 1st- to 3rd-order STSSs help to distinguish the "(a) Moving sth closer to sth" from "(b) Moving sth away from sth", in Figure 5?

3.  Can the format for chart references be standardized? (Figure _vs._ Fig. )

---

> ### Author Response · Authors · 2025-11-21
>
> Thank you for the thorough review. We appreciate your recognition of our well-written presentation, novelty of high-order STSS framework, effective visualizations, and thorough evaluation across multiple tasks. We address your concerns below and will incorporate all clarifications into the revised manuscript during the rebuttal period.
>
> ------
>
>
> > My primary concern revolves around the recency of the compared methods. The main baselines, ATM and Side4Video, were both released in 2023, and generally, most works compared in this paper date from 2023 or earlier. This raises questions about the absence of comparisons with more recent advancements in the field. If there are indeed no significant video understanding (e.g., action recognition) works from 2024 or 2025 that are relevant, it would be important for the authors to clarify this. Otherwise, the lack of comparison with contemporary methods might suggest that incremental modifications to conventional and relatively smaller models may not be considered a substantial contribution to the broader video understanding area.
>
>
> In our main tables, we compare our method with prior approaches focusing on fine-grained motion understanding on motion-centric benchmarks. While several video-related approaches have appeared in 2024-2025, most leverage pretrained LLMs or VLMs to address high-level reasoning or long-video understanding [a1-a4], rather than motion-level understanding in short videos. To the best of our knowledge, there are few recent works on efficient image-to-video transfer in comparable experimental settings.
>
> Nevertheless, fine-grained motion understanding remains crucial for domains such as embodied imitation learning and action quality assessment. Moreover, recent studies [a5, a6] reveal that Video LLMs still lack fine-grained motion understanding capabilities. Our lightweight MOSS module—which parameter-efficiently extracts motion features from pretrained spatial vision transformers—provides primitive temporal cues that can enhance temporal reasoning in contemporary Video LLMs.
>
> To validate this claim and address your concern, **we integrate MOSS with a contemporary Video LLM to verify whether primitive motion modeling enhances comprehensive temporal reasoning**. We use VideoLLaMA3-2B [a7] as our baseline. We extract multi-order STSS features from intermediate visual features of the SigLIP vision encoder through MOSS, and inject them before the projector to provide primitive motion cues to the following LLM. The model is finetuned using LoRA on the publicly available FAVOR-Train set (15K samples) for 1,000 iterations. We evaluate the model on two benchmarks, FAVOR-Bench [a5] and MotionBench [a6], which are designed to evaluate fine-grained temporal reasoning capabilities of Video LLMs.
>
> Tables A and B summarize the results. Compared to the baseline VideoLLaMA 3-2B fine-tuned on the same data, incorporating MOSS improves the overall accuracy by 1.1%p and 2.8%p on FAVOR-Bench and MotionBench, respectively, demonstrating that multi-order STSS features effectively enhance fine-grained motion understanding in Video LLMs. Importantly, this enhancement requires only **13.7M trainable parameters (0.6% out of total parameters) and ~1 hour training on 8×A6000 GPUs**—substantially more efficient than methods [a8-a10] requiring large-scale re-training.
>
> These results demonstrate that our approach serves as an effective foundational component applicable across different paradigms—from efficient transfer learning (our main focus in Tables 1-2) to contemporary Video LLMs. We appreciate this valuable comment, which strengthens our contribution. We will include detailed explanations and full experimental results in the revised manuscript.
>
> |method			|all|AS|CM|HAC|MAD|NSM|SAD|
> |--|--|--|--|--|--|--|--|
> |VideoLLaMA3-2B (zero-shot)	|42.2|43.8|27.3|44.4|48.1|45.3|42.8|
> |VideoLLaMA3-2B (finetuned)	|45.5|44.8|27.8|54.5|51.3|**54.7**|45.3|
> |VideoLLaMA3-2B + MOSS	|**46.6 (+1.1%p)** |**46.8**|**28.8**|**55.0**|**52.2**|53.3|**45.7**|
>
> Table A. **Close-ended evaluation results on FAVOR-Bench**. AS, CM, HAC, MAD, NSM, and SAD denote action sequence, camera motion, holistic action classification, multiple action details, non-subject motion, and single action detail tasks, respectively.
>
> |method			|all|MR|LM|CM|MO|AO|RC|
> |--|--|--|--|--|--|--|--|
> |VideoLLaMA3-2B (zero-shot)	|50.2|54.9|54.2|36.1|68.3|37.0|27.3|
> |VideoLLaMA3-2B (finetuned)	|51.4|55.8|54.4|36.4|**68.8**|38.3|33.0|
> |VideoLLaMA3-2B + MOSS	|**54.2 (+2.8%p)** |**59.7**|**55.7**|**48.3**|68.1|**38.5**|**34.0**|
>
> Table B. **Evaluation results on MotionBench-Dev**. MR, LM, CM, MO, AO, RC denote motion recognition, location-related motion, camera motion, motion-related objects, action order, repetition count tasks, respectively.

---

> ### Author Response · Authors · 2025-11-21
>
> > Regarding the overall architecture, could the authors explain the rationale behind bridging certain layers with MOSS modules while others are connected solely via Fully Connected (FC) layers? Is there any analysis presented to demonstrate that different layers exhibit varying preferences for motion-augmented visual features?
>
> As analyzed in our Appendix (L900-908) and Table 9b, the placement of the MOSS module involves a trade-off between semantic maturity and temporal modeling capacity. Inserting MOSS too early (layer 2) produces noisy STSS due to insufficient visual semantics, while inserting too late (layer 8) limits temporal modeling capacity as fewer blocks remain to process motion features. The middle layers (e.g., after the 4th block) represent a "sweet spot" where features are semantically robust enough for accurate correlation, yet enough subsequent layers remain for effective temporal modeling.
>
> -------------
>
> > Given the paper's analysis that the 2nd order STSS provides motion segment information, and the 3rd order STSS demonstrates the overall layouts of motion segments, it is puzzling why combining the 1st, 2nd, and 3rd order STSS does not lead to better results in Table 4(b). While the authors claim that "redundant temporal dynamics" are captured for the 2nd and 3rd orders, this explanation does not seem to align well with the vivid illustrations provided in the main paper.
>
> Thank you for this insightful question. We provide related discussion in Appendix F. While our theoretical analysis and toy examples (Figure 3) demonstrate that 3rd-order STSS can capture group-wise motion patterns, we empirically observe that the learned model primarily utilizes 3rd-order STSS to capture motion boundaries rather than group dynamics (Figures 5, 8-12). This explains the limited complementarity between 2nd- and 3rd-order STSS in Table 4(b). Inspection of 2nd-order STSS visualizations (Figures 5, 8-12, rows 3) reveals that motion boundary information is already implicitly present in 2nd-order STSS maps—regions where motion segments meet naturally exhibit distinct patterns. Since the learned 3rd-order encoder converges to capturing similar boundary information rather than the group-level dynamics shown in toy examples, combining 2nd+3rd provides limited additional benefit compared to using 2nd-order alone.
>
> -------------
> > How do the 1st- to 3rd-order STSSs help to distinguish the "(a) Moving sth closer to sth" from "(b) Moving sth away from sth", in Figure 5?
>
> We apologize for the unclear visualization in Figure 5. All orders of STSS are capable of representing directional motion information, which enables distinguishing between "closer" and "away" movements. This capability is more clearly demonstrated in other figures (Figures 3, 8a, 8b, and 9b) For instance, in Figure 9b, you can observe that as objects move in a rightward direction, the maps of motion flows, segments, and boundaries at all orders consistently shift rightward, following the object's trajectory. In Figure 5, this directional information is present but subtle in the visualization. We will improve this visualization in the revised manuscript.
>
> -------------
>
> > Can the format for chart references be standardized? (Figure vs. Fig. )
>
> Thanks for your comment. We will standardize the figure reference in the final manuscript during this rebuttal.
>
> -------------
>
> [a1] C. Fu et al. “Video-mme: The first-ever comprehensive evaluation benchmark of multi-modal LLMs in video analysis.”, arXiv 2024. \
> [a2] K. Li et al. “MVBench: A comprehensive multi-modal video understanding benchmark.”, CVPR 2024. \
> [a3] J. Zhou et al. “MLVU: A comprehensive benchmark for multi-task long video understanding.”, arXiv 2024. \
> [a4] B. Wu et al. “Star: A benchmark for situated reasoning in real-world videos.” arXiv 2024. \
> [a5] Tu et al., “FAVOR-Bench: A Comprehensive Benchmark for Fine-Grained Video Motion Understanding,” CoRR, 2025. \
> [a6] Hong et al., “MotionBench: Benchmarking and Improving Fine-grained Video Motion Understanding for Vision Language Models,” CVPR, 2025. \
> [a7] Zhang et al., “VideoLLaMA 3: Frontier Multimodal Foundation Models for Image and Video Understanding,” arXiv, 2025. \
> [a8] Liu et al., “BT-Adapter: Video Conversation is Feasible Without Video Instruction Tuning,” CVPR, 2024. \
> [a9] Nie et al., “SlowFocus: Enhancing Fine-grained Temporal Understanding in Video LLM,” NeurIPS, 2024. \
> [a10] Rasekh et al., “Enhancing Temporal Understanding in Video-LLMs through Stacked Temporal Attention in Vision Encoders,” NeurIPS, 2025.

---

> > ### Comment · Reviewer_LQoE · 2025-11-26
> >
> > Thank you for the detailed responses and explanations. Most of my concerns have been addressed. Given the current situation that "there are few recent works on efficient image-to-video transfer in comparable experimental settings," I am maintaining my rating of 6.

---

> ### Author Response · Authors · 2025-11-27
>
> We appreciate your continued engagement and the decision to maintain the acceptance. Indeed, there have been recent advances in video understanding, including adapting large-scale image/video foundation models (e.g., DINOv2/v3 or V-JEPA 1/2 [a11-a14]) to action recognition and CLIP-based image-to-video transfer with natural language supervision [a15-a17]. Our earlier phrase "few recent works in comparable experimental settings" specifically refers to methods sharing the same backbone (i.e., CLIP) and training supervision (i.e., without text supervision) for fair comparison.
>
> Nevertheless, following your suggestion, we provide a broader comparison with these contemporary methods [a11-a17] in Tables D and E. On Something-Something (Table D), MOSS outperforms image-to-video transfer methods based on image foundation models (DINOv2/v3) [a11, a12] and performs even competitive to video foundation models [a13, a14], despite using significantly fewer parameters (e.g., 386M (ours) vs. 709M (V-JEPA 2)) and no large-scale video pretraining. Additionally, MOSS surpasses text-supervised CLIP-based methods [a15, a16], demonstrating superior motion modeling capabilities without textual guidance. On Kinetics-400 (Table E), MOSS similarly outperforms visual foundation models with substantially smaller model size, and achieves competitive performance with text-supervised methods [a15-a17]. These results demonstrate that our approach remains effective compared to contemporary methods across different paradigms.
>
> We sincerely appreciate your constructive feedback and will incorporate these comparisons in the final manuscript. We hope that these clarifications address your concerns regarding the recency of our work. If you find the broader comparisons strengthen our contribution, we would be grateful if you could consider updating your score.
>
>
> |method|year|frames $\times$ clips| Params (M)| SSV2|
> |--|--|--|--|--|
> DINOv2 ViT-g [a11]			| 2024 | 16 $\times$ 6 |1186| 68.4|
> DINOv3 7B [a12] 			| 2025 | 16 $\times$ 6 | 6756| 70.8 |
> V-JEPA ViT-H [a13] 			| 2024 | 16 $\times$ 6 |658| 74.3|
> V-JEPA 2 ViT-H [a14]			| 2025 | 16 $\times$ 6 |709| 74.0|
> |-|-|-|-|-|-|
> |OmniCLIP-B$^\dagger$ [a15]	|2024| 16 $\times$ 12|n/a|67.3|
> |M$^2$-CLIP-B$^\dagger$ [a16]	|2024|32 $\times$ 3|185|69.1|
> |MOSS-B (ours)			|2025| 16 $\times$ 6|108|72.4|
> |MOSS-L (ours)			|2025| 16 $\times$ 6|386|**74.4**|
>
> Table D. **Comparison with recent methods (2024-) on Something-Something V2**. $\dagger$: trained with natural language supervision.
>
> |method|year|frames $\times$ clips| Params (M)|K400|
> |--|--|--|--|--|
> DINOv2 ViT-g [a11] 				| 2024 | 16 $\times$ 6 |1186| 68.4|
> DINOv3 7B [a12] 			| 2025 | 16 $\times$ 6 |6756| 70.8 |
> V-JEPA ViT-H [a13] 			| 2024 | 16 $\times$ 24 | 658| 84.5|
> V-JEPA 2 ViT-H [a14] 			| 2025 | 16 $\times$ 24 | 709| 85.3|
> |-|-|-|-|-|
> |OmniCLIP-B$^\dagger$ [a15] 	|2024| 8 $\times$ 12|n/a|84.1|
> |M$^2$-CLIP-B$^\dagger$ [a16]	|2024| 16 $\times$ 12|185|83.7|
> |M$^2$-CLIP-B$^\dagger$ [a16]	|2024| 32 $\times$ 12|185|84.1|
> |CLIP4Vis-L$^\dagger$ [a17]		|2025| 8 $\times$ 12 |n/a|87.4|
> |MOSS-B (ours)			|2025| 16 $\times$ 12|108|84.9|
> |MOSS-B (ours)			|2025| 32 $\times$ 12|108|85.2|
> |MOSS-L (ours)			|2025| 16 $\times$ 12|386|**87.7**|
>
> Table E. **Comparison with recent methods (2024-) on Kinetics-400**. $\dagger$: trained with natural language supervision.
>
>
> [a11] Oquab et al., "DINOv2: Learning Robust Visual Features without Supervision," TMLR, 2024. \
> [a12] Siméoni et al., "DINOv3," arXiv, 2025. \
> [a13] Bardes et al., "Revisiting Feature Prediction for Learning Visual Representations from Video," TMLR, 2024. \
> [a14] Assran et al., "V-JEPA 2: Self-Supervised Video Models Enable Understanding, Prediction and Planning," arXiv, 2025. \
> [a15] Liu et al., "OmniCLIP: Adapting CLIP for Video Recognition with Spatial-Temporal Omni-Scale Feature Learning," ECAI, 2024. \
> [a16] Wang et al., "M²-CLIP: A Multimodal, Multi-Task Adapting Framework for Video Action Recognition," AAAI, 2024. \
> [a17] Zheng et al., "Clip4Vis: Parameter-Free Fusion for Multimodal Video Recognition," Neurocomputing, 2025.

---

### Official Review · Reviewer_AMV8 · 2025-10-29

**Soundness:** 2
**Presentation:** 3
**Contribution:** 2
**Rating:** 4
**Confidence:** 3

**Summary:**

This paper introduces the concept of high-order space-time self-similarity (STSS) for video understanding, proposing that while 1st-order STSS captures basic motion flows, 2nd-order STSS identifies coherent motion segments, and 3rd-order STSS captures their layout. To leverage this, the authors created the Multi-Order Self-Similarity (MOSS) module, a lightweight neural component that learns and integrates these multi-order features to enhance motion modeling. When applied to video action recognition, this method achieves new state-of-the-art results on motion-centric datasets like Something-Something V1 & V2, Diving48, and FineGym, demonstrating a superior memory-accuracy trade-off compared to previous approaches.

**Strengths:**

1. The proposed method achieves new state-of-the-art results on several motion-centric action recognition benchmarks, e.g. Something-Something V1 & V2 , Diving48, and FineGym, outperforming previous methods with substantial margins.
2. The paper introduces the Multi-Order Self-Similarity (MOSS) module, which is a lightweight neural module. This module enhances motion modeling capabilities while adding only marginal computation cost and memory usage.

**Weaknesses:**

1. The method is explicitly designed to enhance "motion modeling capabilities" and achieves its state-of-the-art results on motion-centric benchmarks like Something-Something V1/V2, Diving48, and FineGym. However, on benchmarks like Kinetics-400, the performance gain is modest. The MOSS-L model achieves only 0.7% improvement over its baseline. The generalization of such method is yet to be clarified.
2. The paper explores Self-Similarity in videos, in the era of large models, what will be the benefit of such method on MLLM video understanding?
3. The paper's main novelty is the exploration of high-order self-similarity. However, the paper's own analysis shows diminishing returns. While 1st-order STSS provides a significant boost, the individual gains from 2nd, 3rd, and 4th-order STSS become progressively smaller.

**Questions:**

1. Since I am not an expert in this domain, what are the real-world applications of STSS?

---

> ### Author Response · Authors · 2025-11-21
>
> Thank you for the constructive feedback. We appreciate your recognition of our strong performance on motion-centric benchmarks and the lightweight, efficient design of the MOSS module. We carefully address your concerns below and will integrate all discussions into the revised manuscript during the rebuttal period.
>
> -------------
>
> > The method is explicitly designed to enhance "motion modeling capabilities" and achieves its state-of-the-art results on motion-centric benchmarks like Something-Something V1/V2, Diving48, and FineGym. However, on benchmarks like Kinetics-400, the performance gain is modest. The MOSS-L model achieves only 0.7% improvement over its baseline. The generalization of such method is yet to be clarified.
>
> The relatively modest gain on Kinetics-400 stems from the appearance-centric nature of this benchmark, where most action categories (e.g., "motorcycling," "playing guitar," "riding a horse") can easily be recognized solely from static appearance cues such as objects and scenes, without modeling temporal dynamics. Prior studies [a1-a3] have confirmed the marginal contribution of temporal modeling on Kinetics, demonstrating that reversing or permuting video frames leads to minimal accuracy drop on this dataset. Therefore, even when MOSS truly substantially strengthens motion modeling, the benchmark offers limited room to reflect these gains. Consequently, the modest improvement on Kinetics-400 does not indicate limited generalization capability. Rather, the generalizability of our method is demonstrated by consistent and substantial improvements across diverse tasks (e.g., action recognition, action detection, and generic event boundary detection) that require fine-grained temporal modeling (Tables A, B, 1, 7, and 8).
>
> [a1] Xie et al., “Rethinking Spatiotemporal Feature Learning: Speed-Accuracy Trade-offs in Video Classification,” ECCV, 2018. \
> [a2] Neimark et al., “Video Transformer Network,” ICCV, 2021. \
> [a3] Seilla-Lara et al., “Only Time Can Tell:Discovering Temporal Data for Temporal Modeling,” WACV, 2021.

---

> ### Author Response · Authors · 2025-11-21
>
> > The paper explores Self-Similarity in videos, in the era of large models, what will be the benefit of such method on MLLM video understanding?
>
> Thank you for this constructive question. While Video LLMs demonstrate strong capability in story- or event-level temporal reasoning over long videos, recent studies [a4, a5] show they still struggle with fine-grained motion-level reasoning. **Our method can enhance Video LLMs by providing primitive temporal cues for advancing fine-grained motion understanding**.
>
> To validate this, we conduct experiments on two video LLM-related benchmarks, the FAVOR-Bench [a4] and MotionBench [a5], which evaluate fine-grained temporal reasoning capabilities of Video LLMs. We use VideoLLaMA3-2B [a6] as our baseline. We extract multi-order STSS features from intermediate visual features of the SigLIP vision encoder through MOSS, and inject the output features before the projector to provide primitive motion cues to the following LLM. The model is finetuned using LoRA on the publicly available FAVOR-Train set (15K samples) for 1,000 iterations.
>
> Tables A and B summarize the results. Compared to the baseline VideoLLaMA 3-2B fine-tuned on the same data, incorporating MOSS improves the overall accuracy by 1.1%p and 2.8%p on FAVOR-Bench and MotionBench, respectively, demonstrating that multi-order STSS features effectively enhance fine-grained motion understanding in Video LLMs.
>
> Importantly, enhancing the model with MOSS requires only **13.7M trainable parameters (0.6% out of total parameters) and ~1 hour training on 8×A6000 GPUs**, making it substantially more efficient than existing approaches [a9-a11] that require large-scale retraining using massive video datasets. We appreciate this valuable comment, which strengthens our contribution. We will include detailed explanations and full experimental results in the revised manuscript.
>
> |method			|all|AS|CM|HAC|MAD|NSM|SAD|
> |--|--|--|--|--|--|--|--|
> |VideoLLaMA3-2B (zero-shot)	|42.2|43.8|27.3|44.4|48.1|45.3|42.8|
> |VideoLLaMA3-2B (finetuned)	|45.5|44.8|27.8|54.5|51.3|**54.7**|45.3|
> |VideoLLaMA3-2B + MOSS	|**46.6 (+1.1%p)** |**46.8**|**28.8**|**55.0**|**52.2**|53.3|**45.7**|
>
> Table A. **Close-ended evaluation results on FAVOR-Bench**. AS, CM, HAC, MAD, NSM, and SAD denote action sequence, camera motion, holistic action classification, multiple action details, non-subject motion, and single action detail tasks, respectively.
>
> |method			|all|MR|LM|CM|MO|AO|RC|
> |--|--|--|--|--|--|--|--|
> |VideoLLaMA3-2B (zero-shot)	|50.2|54.9|54.2|36.1|68.3|37.0|27.3|
> |VideoLLaMA3-2B (finetuned)	|51.4|55.8|54.4|36.4|**68.8**|38.3|33.0|
> |VideoLLaMA3-2B + MOSS	|**54.2 (+2.8%p)** |**59.7**|**55.7**|**48.3**|68.1|**38.5**|**34.0**|
>
> Table B. **Evaluation results on MotionBench-Dev**. MR, LM, CM, MO, AO, RC denote motion recognition, location-related motion, camera motion, motion-related objects, action order, repetition count tasks, respectively.
>
>
> [a4] Tu et al., “FAVOR-Bench: A Comprehensive Benchmark for Fine-Grained Video Motion Understanding,” CoRR, 2025. \
> [a5] Hong et al., “MotionBench: Benchmarking and Improving Fine-grained Video Motion Understanding for Vision Language Models,” CVPR, 2025. \
> [a6] Zhang et al., “VideoLLaMA 3: Frontier Multimodal Foundation Models for Image and Video Understanding,” arXiv, 2025. \
> [a7] Liu et al., “BT-Adapter: Video Conversation is Feasible Without Video Instruction Tuning,” CVPR, 2024. \
> [a8] Nie et al., “SlowFocus: Enhancing Fine-grained Temporal Understanding in Video LLM,” NeurIPS, 2024. \
> [a9] Rasekh et al., “Enhancing Temporal Understanding in Video-LLMs through Stacked Temporal Attention in Vision Encoders,” NeurIPS, 2025.

---

> ### Author Response · Authors · 2025-11-21
>
> > The paper's main novelty is the exploration of high-order self-similarity. However, the paper's own analysis shows diminishing returns. While 1st-order STSS provides a significant boost, the individual gains from 2nd, 3rd, and 4th-order STSS become progressively smaller.
>
> We view the diminishing returns across orders as a natural outcome of progressive abstraction of temporal dynamics. Physically, 1st-order STSS capturing motion flows implies displacement of the query across frames, which is the most fundamental temporal cue for motion modeling. Higher-order STSSs (motion segments, motion boundaries) offer complementary but less direct information compared to 1st-order motion flows in motion modeling. This mirrors hierarchies in physics, where successive temporal derivatives of position yield progressively distinct physical quantities (position → velocity → acceleration → jerk), each with refined but smaller individual impacts.
>
> Importantly, despite the diminishing returns, we highlight that **high-order STSS still provides consistent and meaningful improvements**. Table 4a shows that each individual order contributes distinct gains: 2nd-order (+1.8%p), 3rd-order (+1.4%p), and 4th-order (+1.0%p). These results validate that each STSS order contributes distinct temporal dynamics beyond those captured by conventional 3D convolutions (+0.6%p) or transformers (0.5%p) in Table 8c. Furthermore, the combination of 1st+2nd orders (Table 4b) achieves 60.0%, clearly surpassing 1st-order alone (59.0%), demonstrating genuine complementarity and validating the utility of high-order STSS beyond the dominant 1st-order STSS.
>
> -------------
>
> > Since I am not an expert in this domain, what are the real-world applications of STSS?
>
> The fundamental value of STSS lies in its ability to capture pure relational structures (e.g., motion dynamics, physical interactions) while suppressing photometric variations (e.g., appearance, illumination, background). This unique property makes it effective for various real-world applications where motion is the primary signal, including motion-based video retrieval [a10] or video motion transfer [a11].
>
> [a10] Shechtman et al., “Matching Local Self-Similarities across Images and Videos,” CVPR, 2007. \
> [a11] Jeong et al., “DreamMotion: Space-Time Self-Similar Score Distillation for Zero-Shot Video Editing,” ECCV, 2024.

---

> > ### Author Response · Authors · 2025-11-28
> >
> > Dear Reviewer AMV8,
> >
> > We greatly appreciate your time and effort in reviewing our paper. Your comments were insightful and helpful, and we have made every effort to address them thoroughly in our rebuttal and the revised manuscript. As the discussion period is coming to a close, we would be grateful if you could let us know whether our responses have sufficiently resolved your concerns. Please don’t hesitate to let us know if anything remains unclear—we would be happy to clarify further. Thank you once again for your thoughtful review and consideration.
> >
> > Best regards, \
> > The Authors

---

### Official Review · Reviewer_LYWe · 2025-11-03

**Soundness:** 3
**Presentation:** 3
**Contribution:** 2
**Rating:** 4
**Confidence:** 3

**Summary:**

This paper presents a method of high-order space-time self-similarities (STSS) for video understanding. It consists of stacks of self-similarity modules that are similar to self-correlation modules where the difference is general similarity vs. vanilla correlation. The authors then stacks multiple layers of STSS in the neural network model. The authors argue that the proposed method achieves SOTA results on several motion-centric video understanding benchmarks.

**Strengths:**

- The proposed method achieves SOTA on multiple motion-centric benchmarks.
- Extensive experiments and ablation studies are provided. The analysis and visualizations provide good insights.
- The presentation of this paper is good and easy to understand and follow.
- The toy visualization in Figure 3 is a good way to help readers understand the method.

**Weaknesses:**

- The novelty is somewhat limited. The proposed STSS superficially looks interesting. However, the core operation is essentially multiple stacks of correlation/distance + encoding. I believe there has been multiple previous works with similar ideas. It would be surprising to me if the proposed method achieve SOTA while previous methods did not.
- In Table 4, it seems that higher order of STSS does not necessarily lead to increased performance. In fact, the authors admits that "STSS beyond 3rd-order do not provide significant benefits for action recognition tasks". This may have shown that high orders of STSS may not be necessary or even correct, which makes the contribution of this paper quite limited.
- The proposed method heavily depends on the pre-trained encoder as shown in Table 9. Not sure if this is OK.
- Not sure how the proposed STSS module can be used in other video related operations such as generative models.

**Questions:**

- What is the difference between the proposed STSS vs. attention mechanism in transformer? If the similarity function $\phi$ is replaced by a learnable similarity, isn't it be reduced to an attention layer whose attention mask applies to the rectangular region of $L\times U\times V$?
- How can STSS module be used in other video related operations such as diffusion models?

---

> ### Author Response · Authors · 2025-11-21
>
> Thank you for the thoughtful review. We appreciate your recognition that our experimental analysis extensive and insightful, our visualizations helpful, and our presentation clear and easy to follow. We address your concerns and questions below and will incorporate all clarifications into the revised manuscript during the rebuttal period.
>
> -------------
>
> > The novelty is somewhat limited. The proposed STSS superficially looks interesting. However, the core operation is essentially multiple stacks of correlation/distance + encoding. I believe there has been multiple previous works with similar ideas. It would be surprising to me if the proposed method achieve SOTA while previous methods did not.
>
> We respectfully clarify that our core contribution lies in **introducing the novel concept of high-order spatio-temporal self-similarity** and providing in-depth analysis **demonstrating that hierarchically distinct temporal abstractions emerge at each order**. Specifically, we provide: (i) conceptual and qualitative analysis identifying what temporal dynamics emerge at each order (Section 3.2, Figures 3, 5, 8-12), (ii) quantitative analysis of individual effects and cross-order complementarity (Tables 4a-4b, Figure 7), and (iii) design principles for maximizing this complementarity (Table 4c). **As the reviewer acknowledges, "the analysis and visualizations provide good insights"—this in-depth analysis of higher-order STSS fundamentally distinguishes our work from prior approaches [a1, a2] that remained limited to 1st-order STSS**. This explains how our method achieves stronger performance with lower computational cost (Table 9c). The "stack of correlation+encoding blocks" is merely the implementation of our analytical findings, not our contribution itself.
>
> [a1] Kwon et al., “Learning Self-Similarity in Space and Time as Generalized Motion for Video Action Recognition,” ICCV, 2021. \
> [a2] Wu et al., “What Can Simple Arithmetic Operations Do for Temporal Modeling?,” ICCV, 2023.
>
> -------------
>
>
> > In Table 4, it seems that higher order of STSS does not necessarily lead to increased performance. In fact, the authors admit that "STSS beyond 3rd-order do not provide significant benefits for action recognition tasks". This may have shown that high orders of STSS may not be necessary or even correct, which makes the contribution of this paper quite limited.
>
> Thank you for the thoughtful comment. We first clarify that each high-order STSS does provide consistent performance improvements: in Table 4a, 2nd-order (+1.8%p), 3rd-order (+1.4%p), and 4th-order (+1.0%p) all outperform the baseline (56.9%). These results validate that each STSS order contributes distinct temporal dynamics beyond those captured by conventional 3D convolutions (+0.6%p) or transformers (0.5%p) in Table 8c. **The diminishing gains and marginal improvements beyond 3rd-order ($N>3$) do not indicate that high-order STSS is unnecessary or incorrect—rather, this is a natural outcome of progressive abstraction**. Physically, 1st-order STSS capturing motion flows implies displacement of the query across frames, which is indeed the most fundamental temporal cue for motion modeling. Higher orders (motion segments, boundaries) offer complementary but progressively more abstract information, naturally yielding smaller individual contributions. This mirrors hierarchies in physics, where successive temporal derivatives of position yield progressively distinct physical quantities (position → velocity → acceleration → jerk), each with refined but smaller individual impacts.
>
> -------------
>
> > The proposed method heavily depends on the pre-trained encoder as shown in Table 9. Not sure if this is OK.
>
> We are not fully certain which aspect the reviewer believes our method "depends" on regarding the pretrained encoder. The key takeaway from Table 9(e) is **the agnosticity of our method to the pretrained encoder**: high-order STSS consistently provides substantial performance gains across diverse backbones (CLIP, DINO, MAE), rather than relying on a specific encoder. We would like to clarify that the **variations in absolute performance across encoders are natural** since each pretrained encoder exhibits different levels of generalizability depending on its pretraining objective and data [a3-a5].
>
> [a3] Ericsson et al., “How Well Do Self-supervised Models Transfer?” CVPR, 2021. \
> [a4] Yant et al., “Adapting Image Models for Efficient Video Action Recognition,” ICLR, 2023. \
> [a5] Park et al., “Dual-path Adaptation from Image to Video Transformers,” CVPR, 2023.

---

> ### Author Response · Authors · 2025-11-21
>
> > Not sure how the proposed STSS module can be used in other video related operations such as generative models.
>
> > How can STSS module be used in other video related operations such as diffusion models?
>
> Thank you for this constructive suggestion. While exploring STSS in video generation is an interesting direction, we believe direct integration of MOSS module with video diffusion models is not straightforward. Since diffusion models progressively generate videos from random noise, STSS computed during intermediate denoising steps would likely be noisy and unreliable, leading to inaccurate motion features that may degrade generation quality.
>
> **Instead, we explore the applicability of MOSS to Video LLMs for fine-grained motion understanding**. While Video LLMs demonstrate strong capability in story- or event-level temporal reasoning over long videos, recent studies [a6, a7] show they still struggle with fine-grained motion-level reasoning. **Our method can enhance Video LLMs by providing primitive temporal cues for advancing fine-grained motion understanding**.
>
> To validate this, we conduct experiments on two video LLM-related benchmarks, the FAVOR-Bench [a6] and MotionBench [a7], which evaluate fine-grained temporal reasoning capabilities of Video LLMs. We use VideoLLaMA3-2B [a8] as our baseline. We extract multi-order STSS features from intermediate visual features of the SigLIP vision encoder through MOSS, and inject the output features before the projector to provide primitive motion cues to the following LLM. The model is finetuned using LoRA on the publicly available FAVOR-Train set (15K samples) for 1,000 iterations.
>
> Tables A and B summarize the results. Compared to the baseline VideoLLaMA 3-2B fine-tuned on the same data, incorporating MOSS improves the overall accuracy by 1.1%p and 2.8%p on FAVOR-Bench and MotionBench, respectively, demonstrating that multi-order STSS features effectively enhance fine-grained motion understanding in Video LLMs.
>
> Notably, enhancing the model with MOSS requires only **13.7M trainable parameters (0.6% out of total parameters) and ~1 hour training on 8×A6000 GPUs**, making it substantially more efficient than existing approaches [a9-a11] that require large-scale retraining using massive video datasets. We appreciate this valuable comment, which strengthens our contribution. We will include detailed explanations and full experimental results in the revised manuscript.
>
> |method			|all|AS|CM|HAC|MAD|NSM|SAD|
> |--|--|--|--|--|--|--|--|
> |VideoLLaMA3-2B (zero-shot)	|42.2|43.8|27.3|44.4|48.1|45.3|42.8|
> |VideoLLaMA3-2B (finetuned)	|45.5|44.8|27.8|54.5|51.3|**54.7**|45.3|
> |VideoLLaMA3-2B + MOSS	|**46.6 (+1.1%p)** |**46.8**|**28.8**|**55.0**|**52.2**|53.3|**45.7**|
>
> Table A. **Close-ended evaluation results on FAVOR-Bench**. AS, CM, HAC, MAD, NSM, and SAD denote action sequence, camera motion, holistic action classification, multiple action details, non-subject motion, and single action detail tasks, respectively.
>
> |method			|all|MR|LM|CM|MO|AO|RC|
> |--|--|--|--|--|--|--|--|
> |VideoLLaMA3-2B (zero-shot)	|50.2|54.9|54.2|36.1|68.3|37.0|27.3|
> |VideoLLaMA3-2B (finetuned)	|51.4|55.8|54.4|36.4|**68.8**|38.3|33.0|
> |VideoLLaMA3-2B + MOSS	|**54.2 (+2.8%p)**|**59.7**|**55.7**|**48.3**|68.1|**38.5**|**34.0**|
>
> Table B. **Evaluation results on MotionBench-Dev**. MR, LM, CM, MO, AO, RC denote motion recognition, location-related motion, camera motion, motion-related objects, action order, repetition count tasks, respectively.
>
> [a6] Tu et al., “FAVOR-Bench: A Comprehensive Benchmark for Fine-Grained Video Motion Understanding,” CoRR, 2025. \
> [a7] Hong et al., “MotionBench: Benchmarking and Improving Fine-grained Video Motion Understanding for Vision Language Models,” CVPR, 2025. \
> [a8] Zhang et al., “VideoLLaMA 3: Frontier Multimodal Foundation Models for Image and Video Understanding,” arXiv, 2025. \
> [a9] Liu et al., “BT-Adapter: Video Conversation is Feasible Without Video Instruction Tuning,” CVPR, 2024. \
> [a10] Nie et al., “SlowFocus: Enhancing Fine-grained Temporal Understanding in Video LLM,” NeurIPS, 2024. \
> [a11] Rasekh et al., “Enhancing Temporal Understanding in Video-LLMs through Stacked Temporal Attention in Vision Encoders,” NeurIPS, 2025.

---

> ### Author Response · Authors · 2025-11-21
>
> > What is the difference between the proposed STSS vs. attention mechanism in transformer? If the similarity function  is replaced by a learnable similarity, isn't it be reduced to an attention layer whose attention mask applies to the rectangular region of (L, U, V)?
>
> Thank you for this valuable question. As the reviewer correctly points out, the query-key similarity tensor in self-attention can be viewed as 1st-order STSS with minor modifications. However, the fundamental difference lies in how the similarity tensor is utilized.
>
> Self-attention breaks down the STSS tensor into individual query-key pair-wise similarities and uses each similarity score as a weight for aggregating corresponding value elements. This decomposition makes the model blind to the holistic relational patterns within the STSS tensor, leading to the permutation-invariant output, limiting its ability to effectively capture motion patterns. **In contrast, our method directly encodes the entire STSS tensor as a whole, preserving spatio-temporal relational structures and explicitly transforming them into motion features**.
>
> To validate this distinction, we compare MOSS with local self-attention using the same (L,U,V) extent (Table C). While local self-attention provides marginal improvement (+0.5%p), directly encoding the STSS tensor yields significantly larger gains (+3.1%p), confirming that using entire relational patterns in STSS as motion is crucial for effective temporal modeling. This finding aligns well with prior work [a1] demonstrating the benefits of explicit STSS encoding over attention-style aggregation.
>
> |method | SSV1 | D48|
> |--|--|--|
> |baseline | 56.9 | 85.0|
> |+ attention| 57.5|85.5|
> |+ single STSS encoder|59.0|86.3|
> |+ two STSS encoders |**60.0**|**87.7**|
>
> Table C. **Comparison between local self-attention and STSS encoding**. Both use the same spatio-temporal extent (L, U, V)=(5, 9, 9).
>
> [a1] Kwon et al., “Learning Self-Similarity in Space and Time as Generalized Motion for Video Action Recognition,” ICCV, 2021.

---

> > ### Author Response · Authors · 2025-11-28
> >
> > Dear Reviewer LYWe,
> >
> > We deeply appreciate the thoughtful and constructive feedback you provided during the review process. Your thoughtful feedback has been instrumental in helping us refine and clarify our work. We have carefully considered and addressed each of your comments in our rebuttal and the revised manuscript. As the discussion phase nears its end, we would sincerely appreciate it if you could confirm whether our responses have resolved your concerns. If anything remains unclear, we are more than happy to offer additional clarification.
> >
> > Best regards, \
> > The Authors

---

### Author Response · Authors · 2025-11-27
**Revised Manuscript Uploaded**

We sincerely thank all reviewers for their constructive comments. We appreciate the recognition of the novelty of high-order STSS (LQoE), the effectiveness and efficiency of the MOSS module (AMV8), the thorough validation across multiple benchmarks and tasks (LYWe, AMV8, LQoE), the clear presentation with insightful analysis and visualizations (LYWe, LQoE).

Following your valuable suggestions, we have revised and uploaded the manuscript accordingly. We have marked the updated parts in blue to aid in your review. The main updates include:
- Integration with Video-LLMs (Sec. 4.5 & Tab. 5): We integrated MOSS with VideoLLaMA3 to demonstrate its broader applicability to contemporary video understanding models.
- Broader Comparison (Sec. 4.2 & Tabs. 1-2): We incorporated additional references and provided broader comparisons against recent image-to-video transfer methods and video foundation models.
- Comparison with Local Self-Attention (Sec. C.3 & Tab. 10c): We added an experimental comparison between conventional local self-attention and our STSS encoding to validate our design choices.
- Enhanced Visualization (Fig. 5b): We updated the visualizations to more clearly illustrate the temporal evolution of the 1st- to 3rd-order STSS maps.

We believe the paper has been significantly improved thanks to the reviewers' valuable feedback. We would be grateful for any additional comments or questions, and are happy to address them during this rebuttal period.

---

### Meta-Review · Area_Chair_mmJb · 2026-01-03

**Summary:**

The paper proposes a novel Spatio-Temporal Self-Similarity (STSS) module for video understanding. While the idea of exploring high-order self-similarity is interesting and the reported results on specific benchmarks are promising, the manuscript in its current form faces significant challenges regarding ‌novelty justification, generalizability, and contemporary relevance‌. These issues collectively weaken the perceived impact of the contribution.

**1. ‌Insufficient Justification of Novelty and Contribution:**

**(1)** R1 directly questions the novelty, suggesting the core operation resembles established techniques. The authors must provide a much clearer and more rigorous discussion differentiating STSS from prior works on correlation, self-similarity, and attention mechanisms. A detailed conceptual and mathematical comparison with the Transformer attention mechanism (as asked by R1) is essential.

**(2)** The observation of ‌diminishing returns‌ from higher-order STSS (R1, R2) inadvertently undermines the paper's central premise. If gains saturate at the 3rd order, the contribution of proposing a generalized high-order framework is significantly reduced. The authors need to reframe the contribution, perhaps positioning it as identifying the sufficient order for effective motion modeling rather than proposing an endlessly scalable one.

‌**2. Limited Scope and Generalizability‌:**

**(1)** The performance on ‌Kinetics-400‌ is a critical data point. The modest gain suggests the method may be over-specialized for "motion-centric" tasks and less effective for broader "scene-centric" video understanding (R2). The paper must discuss this dichotomy and honestly address the limitations in generalizability.

**(2)** Questions about application to ‌other video domains‌ (generative models, diffusion models, MLLMs) (R1, R2) remain unanswered. While extending to every domain is not required, a discussion on the potential applicability or fundamental constraints of the STSS idea would strengthen the paper's scope and inspire future work.

‌**3. Outdated Benchmarking and Missing Analysis‌:**

**(1)** R3's concern about ‌outdated comparisons‌ is very serious for a rapidly evolving field. The authors must either include comparisons with state-of-the-art methods from 2024-2025 or provide a compelling justification for their absence (e.g., code/data unavailability, focusing on a specific architectural family). Failing to do so makes it difficult to assess the true competitive standing of the proposed method.

**(2)** The ‌architectural design choice‌ (MOSS vs. FC connections) appears ad-hoc without principled justification or analysis (R3). The authors should provide an ablation or analysis (e.g., visualizing feature sensitivities at different layers) to explain this design.

Although the foundation of this work is promising, the manuscript requires significant strengthening in its narrative, analysis, and positioning within the current research landscape.

**Reviewer Concerns:**

1. The main concern on diminishing returns‌ from higher-order STSS (R1, R2) was not addressed by the authors' responses. In Table 4(b) , "1st+2nd order" leads to 1% gain over "1st order only" on SSV1, but "1st +2nd+3rd order" leads to 0.7% drop over "1st+2nd order".

2. The main concern on the modest gain on ‌Kinetics-400‌ (R2) remained unsolved. This suggests that the method may be over-specialized for "motion-centric" tasks and less effective for broader "scene-centric" video understanding.

3. The main concerns of R3 were addressed by the rebuttal. This reviewer had stated to maintain the positive rating.

**Reviewer Scores:**

Since the main concerns of R1 and R2 remained unsolved (both gave negative scores), I tend to recommend a reject for this paper.

---

### Decision · Program_Chairs · 2026-01-26

Reject